RESEARCH

# Dynamics of alternative splicing during somatic cell reprogramming reveals functions for RNA-binding proteins CPSF3, hnRNP UL1, and TIA1

Claudia Vivori[1,2,3], Panagiotis Papasaikas[1,4], Ralph Stadhouders[1,5], Bruno Di Stefano[1,6], Anna Ribó Rubio[1], Clara Berenguer Balaguer[1,7], Serena Generoso[1,2], Anna Mallol[1], José Luis Sardina[1,7], Bernhard Payer[1,2], Thomas Graf[1,2] and Juan Valcárcel[1,2,8*]

* Correspondence: juan.valcarcel@crg.eu
[1]Centre for Genomic Regulation (CRG), The Barcelona Institute of Science and Technology, Carrer del Dr. Aiguader 88, 08003 Barcelona, Spain
[2]Universitat Pompeu Fabra (UPF), Carrer del Dr. Aiguader 88, 08003 Barcelona, Spain
Full list of author information is available at the end of the article

## Abstract

**Background:** Somatic cell reprogramming is the process that allows differentiated cells to revert to a pluripotent state. In contrast to the extensively studied rewiring of epigenetic and transcriptional programs required for reprogramming, the dynamics of post-transcriptional changes and their associated regulatory mechanisms remain poorly understood. Here we study the dynamics of alternative splicing changes occurring during efficient reprogramming of mouse B cells into induced pluripotent stem (iPS) cells and compare them to those occurring during reprogramming of mouse embryonic fibroblasts.

**Results:** We observe a significant overlap between alternative splicing changes detected in the two reprogramming systems, which are generally uncoupled from changes in transcriptional levels. Correlation between gene expression of potential regulators and specific clusters of alternative splicing changes enables the identification and subsequent validation of CPSF3 and hnRNP UL1 as facilitators, and TIA1 as repressor of mouse embryonic fibroblasts reprogramming. We further find that these RNA-binding proteins control partially overlapping programs of splicing regulation, involving genes relevant for developmental and morphogenetic processes.

**Conclusions:** Our results reveal common programs of splicing regulation during reprogramming of different cell types and identify three novel regulators of this process and their targets.

**Keywords:** Alternative splicing, Somatic cell reprogramming, CPSF3, hnRNP UL1, TIA1, Pluripotency

## Background

Alternative splicing (AS) is a widespread mechanism of gene regulation that generates multiple mRNA isoforms from a single gene, dramatically diversifying the transcriptome (and proteome) of eukaryotic cells. Ninety-five percent of multi-exonic mammalian genes undergo AS, producing mRNA isoforms which often differ in coding capacity, stability, or translational efficiency and that can be translated into proteins with distinct structural and functional properties [1, 2]. AS contributes to the regulation of many biological processes in multicellular eukaryotes, including embryonic development and tissue specification (reviewed in [3]). During the last decade, progress has been made to understand the role of post-transcriptional regulation (including AS) in the maintenance of cellular pluripotency and cell fate decisions, discovering genes differentially spliced between stem cells and differentiated cells and splicing regulators that control these choices [4–11].

During reprogramming into induced pluripotent stem (iPS) cells, somatic cells revert to a pluripotent state after overexpression of the transcription factors OCT4, SOX2, KLF4, and MYC (OSKM) [12]. Substantial progress has been made to understand the process at the transcriptional and epigenetic level, such as by identifying numerous roadblocks and some facilitators, but comparatively little is known about how post-transcriptional regulation impacts cell fate decisions. Recent work has revealed the functional relevance and conservation of splicing regulation during reprogramming [7, 13–16]. A conserved functional splicing program associated with pluripotency and repressed in differentiated cells by the RNA-binding proteins MBNL1 and MBNL2 was previously reported [7]. This splicing program includes a mutually exclusive exon event in the transcription factor FOXP1: a switch in inclusion of *Foxp1* exons 16/16b modulates the functions of the transcription factor between pluripotent and differentiated cells [6]. Illustrating the complexity of such splicing program, dynamic changes of AS during cell reprogramming of mouse embryonic fibroblasts (MEFs) revealed sequential waves of exon inclusion and skipping in reprogramming intermediates and the functional role of splicing regulators in modulating reprogramming, in particular during the initial mesenchymal-to-epithelial transition (MET) phase [17]. Given the limited efficiency of cell reprogramming in this system, subpopulations of reprogramming intermediates had to be isolated through the expression of a pluripotency marker, biasing studies to the most prevalent and dominant factors.

Here we took advantage of the rapid, highly efficient and largely synchronous reprogramming of pre-B cells (hereafter referred to as "B cell reprogramming"), obtained by a pulse of the transcription factor C/EBPα followed by induced OSKM expression [18, 19] to study the dynamics of AS during this transition. The essentially homogeneous reprogramming of the cells in this system allowed detailed temporal transcriptome analyses of the bulk population, without the need of selecting for reprogramming intermediates. We established clusters of temporal regulation and compared these changes with the ones differentially spliced in MEF reprogramming [17]. Analyzing the dynamic expression of RNA-binding proteins (RBPs) during reprogramming, we inferred potential AS regulators, of which three were studied in detail. Characterization of these factors, namely CPSF3, hnRNP UL1, and TIA1, by perturbation experiments demonstrated their role as AS regulators in the induction of pluripotency.

## Results

### Dynamics of alternative splicing in C/EBPα-enhanced B cell reprogramming occur independently from gene expression changes

To study the dynamics of changes in alternative splicing (AS) during cell reprogramming, primary mouse pre-B cells (hereafter referred to as "B cells") were reprogrammed as previously described [18–20]. Briefly, B cells were isolated from bone marrow of reprogrammable mice, containing a doxycycline-inducible OSKM cassette and an OCT4-GFP reporter. These cells were infected with a retroviral construct containing an inducible version of C/EBPα fused to the estrogen receptor ligand-binding domain (ER). Infected cells were selected and re-plated on a feeder layer of inactivated MEFs. This was followed by a 18 h-long pulse of β-estradiol, triggering the translocation of C/EBPα-ER to the nucleus and poising the B cells for efficient and homogeneous reprogramming [18, 19]. After washout of β-estradiol, reprogramming was induced by growing the cells for 8 days in reprogramming medium containing doxycycline (see "Methods" and [20]). RNA was isolated every other day from duplicates and subjected to paired-end sequencing (RNA-seq), resulting in high coverage (more than 100 million reads per sample) (Fig. 1A). As positive controls, mouse embryonic stem (ES) cells and induced pluripotent stem (iPS) cells were also sequenced.

AS analysis was performed using *vast-tools* [21], an event-based software that quantifies percent spliced in (PSI) values of annotated AS events and constitutive exons in all samples. These analyses revealed more than 14,000 AS changes during the entire reprogramming time course (for any possible pair of conditions: minimum absolute ΔPSI of 10 between PSI averages and minimum difference of 5 between any individual replicates across conditions) and a gradual increase in the number of differentially spliced events when comparing B cells to progressive stages of somatic cell reprogramming (Fig. 1B and Additional file 1: Figure S1A). Different classes of AS events were detected, with similar relative proportions at the various time points: 31–47% cassette exons, 5–11% alternative 3′ splice sites and 6–11% alternative 5′ splice sites, and 33–57% retained introns (Fig. 1B).

To classify the dynamics of AS changes during reprogramming, we selected cassette exon (CEx) events differentially spliced in at least one comparison (4556 exons) and performed a fuzzy c-means clustering analysis on their scaled PSI values (*Mfuzz*; [22]). This analysis revealed diverse kinetics of exon inclusion occurring during B cell reprogramming (Fig. 1C, Additional file 1: Figure S1B-C and Additional file 2: Table S1). Six major clusters of AS dynamic profiles were detected: (1) exons that become included already after the C/EBPα pulse; (2) and (3) exons that are regulated either towards increased inclusion or skipping early after OSKM induction (day 2); (4) and (5) exons that display changes in inclusion at middle stages of reprogramming (day 4 onwards); (6) a cluster of exons only included at the latest steps of reprogramming (day 8 onwards) (Fig. 1C). Each of these clusters consisted of 300–500 exons. The inclusion levels of examples of exons belonging to different cluster types were validated by semi-quantitative RT-PCR (Fig. 1D). For reference, changes in the PSI values of *Grhl1* exon 6 and *Dnmt3B* exon 10, previously described to be associated with reprogramming and pluripotency [17, 23, 24] were also quantified and found to follow similar inclusion

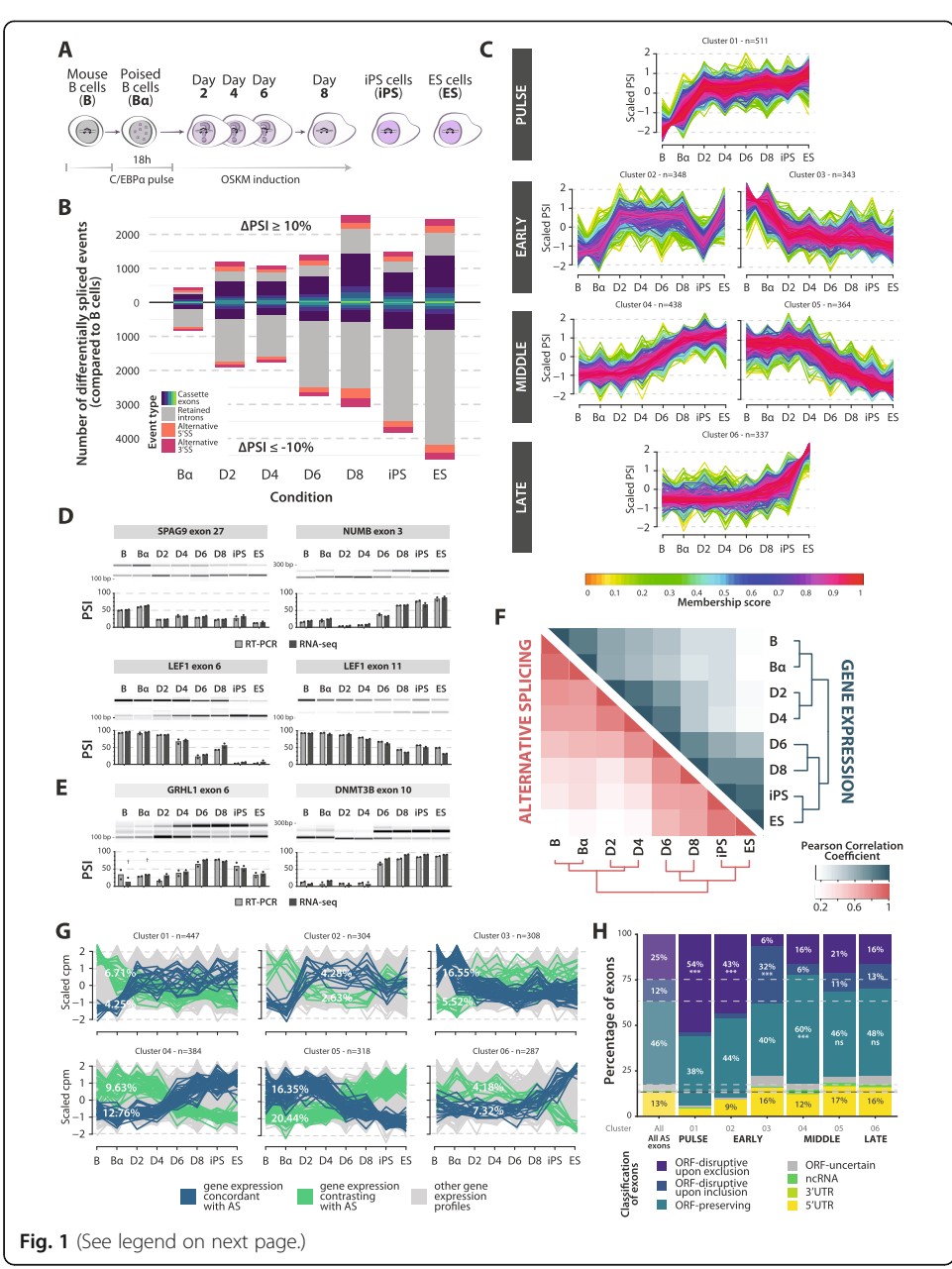

**Fig. 1** (See legend on next page.)

(See figure on previous page.)

**Fig. 1** Dynamics of alternative splicing and gene expression changes during B cell reprogramming. **A** Schematic representation of C/EBPα-mediated B cell reprogramming time points and related controls analyzed by RNA-seq. Bα cells: B cells treated for 18 h with β-estradiol to activate C/EBPα. **B** Stacked bar plot representing cumulative number of events differentially spliced between B cells and subsequent reprogramming stages (x-axis), as well as controls (iPS and ES cells). The y-axis represents the number of differentially spliced events compared to B cells. The upper part corresponds to events with positive ΔPSI values compared to B cells (> 10%), the lower part to events with negative ΔPSI values compared to B cells (< − 10%). Red/orange areas: alternative 3′/5′ splice sites (Alt3/Alt5) respectively; grey areas: retained introns; blue-green areas: cassette exons of increasing complexity (see "Methods"). See also Additional file 1: Figure S1A. **C** Clusters of cassette exons displaying related profiles of inclusion level changes during B cell reprogramming. Six clusters (out of a total of 12 identified, see Additional file 1: Figure S1B-C) are shown and classified into 4 categories, corresponding to the timing of the main shift observed (left). The y-axis represents scaled percent spliced in (PSI) values. The color of each line corresponds to the *membership* score of each exon relative to the general trend of the cluster. The size (number of events) of each cluster is indicated (n). **D** RT-PCR validation of selected cassette exon changes inferred from RNA-seq analyses. AS events assigned to different clusters were analyzed by semi-quantitative RT-PCR and quantified by capillary electrophoresis. Each panel includes a gel representation of the inclusion (top) and skipping (bottom) amplification products of one of the replicates and the corresponding quantification of the duplicates (PSI = molarity of inclusion product / molarity of inclusion + skipping products). Light grey columns: PSI values quantified by RT-PCR; dark grey columns: PSI values quantified by RNA-seq analysis using *vast-tools* software (n = 2). **E** Validation of *Grhl1* exon 6 and *Dnmt3b* exon 10 inclusion level changes, previously associated with reprogramming and pluripotency, performed as in panel **D**. Crosses indicate time points for which PSI values were calculated with low coverage (less than 10 actual reads). **F** Heatmap displaying correlations between B cell reprogramming stages according to gene expression (blue, top heatmap) and AS (red, bottom heatmap). Color scales represent Pearson correlation coefficient values calculated on the cpm values of the 25% most variably expressed genes or upon the PSI values of the 25% most variable cassette exons. **G** Gene expression patterns of genes containing the exons belonging to each of the AS clusters in panel **C**. Genes with expression changes correlating with the cluster centroid or its negative (membership > 0.3) are highlighted in blue and green, respectively, while the grey portion represents the (majority of) genes which follow gene expression profiles that do not match the changes in inclusion patterns of their exon(s). Percentage of concordant/contrasting patterns are displayed for each cluster. See also Additional file 1: Figure S1D. **H** Stacked bar plot representing the percentage of cassette exons in each of the AS clusters in panel **C** classified according to the following categories: disrupting the open reading frame (ORF) upon exclusion or inclusion, preserving the transcript ORF, mapping in non-coding RNAs, 3′ UTRs or 5′ UTRs or uncertain. The first column represents all exons differentially spliced between at least one pair of conditions, while the following ones represent the exons belonging to the each AS cluster (indicated below the bar). Fractions < 5% are not indicated. Where indicated, statistical significance was calculated by Fisher's exact test on the proportion of exons in the cluster compared to the general distribution of all AS exons (*, **, *** = p value < 0.05, 0.01, 0.001 respectively)

patterns in B cell reprogramming, compared to the ones previously described in other systems (Fig. 1E).

We next sought to compare general AS and gene expression dynamics during reprogramming. Gene expression was analyzed using the *edgeR* package [25] and the level of similarity between each pair of conditions was estimated using a Pearson correlation coefficient on the cpm (counts per million) values of the most variable genes (3rd quartile coefficient of variation, $n = 2961$). In addition, Pearson correlation coefficient was calculated based on the PSI values from the most variable cassette exons (3rd quartile coefficient of variation, $n = 1140$). Both analyses showed pronounced switches between days 4 and 6 post-OSKM induction (Fig. 1F). Overall, however, most clusters displayed matching profiles in gene expression and AS of any included exon in less than 10% of the genes, reaching a maximum of 20% in a subset of AS clusters (Fig. 1G and Additional file 1: Figures S1D-E). These results argue, as observed before in a variety of other systems (e.g., [26]), that global programs of regulation of gene expression and AS are uncoupled from each other.

Interestingly, a larger proportion of exons included (or skipped) at early stages of reprogramming are predicted to disrupt the open reading frame (ORF) of the transcripts upon exon skipping (or inclusion, respectively), compared to middle/late exons and to

the general distribution of mapped alternative exons (Fig. 1H, classification as described in [21]). This suggests a higher impact of AS-mediated on/off regulation of the corresponding protein expression via nonsense-mediated decay (NMD), and a switch to expression of full-length proteins during early steps of reprogramming. Middle clusters, instead, contain more exons predicted to preserve the coding potential of their transcripts, implying modulation of the functions of their encoded protein isoforms rather than on/off switches (Fig. 1H). Consistent with these concepts, while PSI values of cassette exons tend to increase throughout reprogramming (Fig. 1B, blue bars), intron retention—generally leading to NMD—tends to decrease in the course of reprogramming (Fig. 1B, grey bars).

### AS changes at intermediate reprogramming stages show commonalities with MEF reprogramming

As a first step to identify key AS events and potential regulators important for reprogramming, we compared our transcriptome analysis of B cell reprogramming with that of MEF reprogramming [17]. The two datasets differ in the experimental design and time frame (compare Figs. 1A and 2A). Specifically, the MEF system required sorting of cells undergoing reprogramming using the SSEA1 surface marker, while this was not necessary for B cell reprogramming due to its efficiency. To facilitate the comparison between the two transcriptome datasets, *vast-tools* analysis was applied to the MEF dataset [17], which yielded 843 cassette exons differentially spliced ($|\Delta PSI| \geq 10$, range $\geq 5$) between any pair of samples of the MEF reprogramming dataset. Despite differences in the experimental setup, 79% of these exons (669 out of 843) were also found to be differentially spliced in B cell reprogramming (Additional file 1: Figure S2A and Additional file 3: Table S2). A similar level of overlap was also observed at the level of other AS events (72% of AS events in general, Additional file 1: Figure S2A). The overlap between differentially spliced exons in the two systems was higher for middle and late AS clusters than for early clusters of B cell reprogramming (Fig. 2B), as might be expected from the convergence on a common program of AS related to pluripotency.

To further compare the two datasets, we performed a Principal Component Analysis (PCA) on the gene expression of the most variable genes in both datasets (3rd quartile coefficient of variation, $n = 2679$) separating the stages into four distinct groups by $k$-means clustering (Fig. 2C). These groups clearly distinguish between starting cells, early and late stages of reprogramming and pluripotent cells. The PCA allowed us to outline a "reprogramming pseudotime" which was subsequently used to select AS exons and regulators for functional characterization. It also further highlighted the transition between days 4 and 6 in B cell reprogramming, juxtaposing them with days 7/10 in MEF reprogramming. A heatmap displaying the scaled PSI values of the 669 common differentially spliced exons of the two datasets at the different steps of the reprogramming process shows substantial similarities in exon inclusion (Fig. 2D).

In contrast to the similarities in AS profiles between the two reprogramming systems, we observed that expression of RNA-binding proteins (RBPs) previously associated with pluripotency, somatic cell reprogramming, and/or development [7, 9, 14, 17] and presumably mechanistically linked to different aspects of post-transcriptional regulation, differed significantly between the two datasets (Fig. 2E). Despite these more divergent

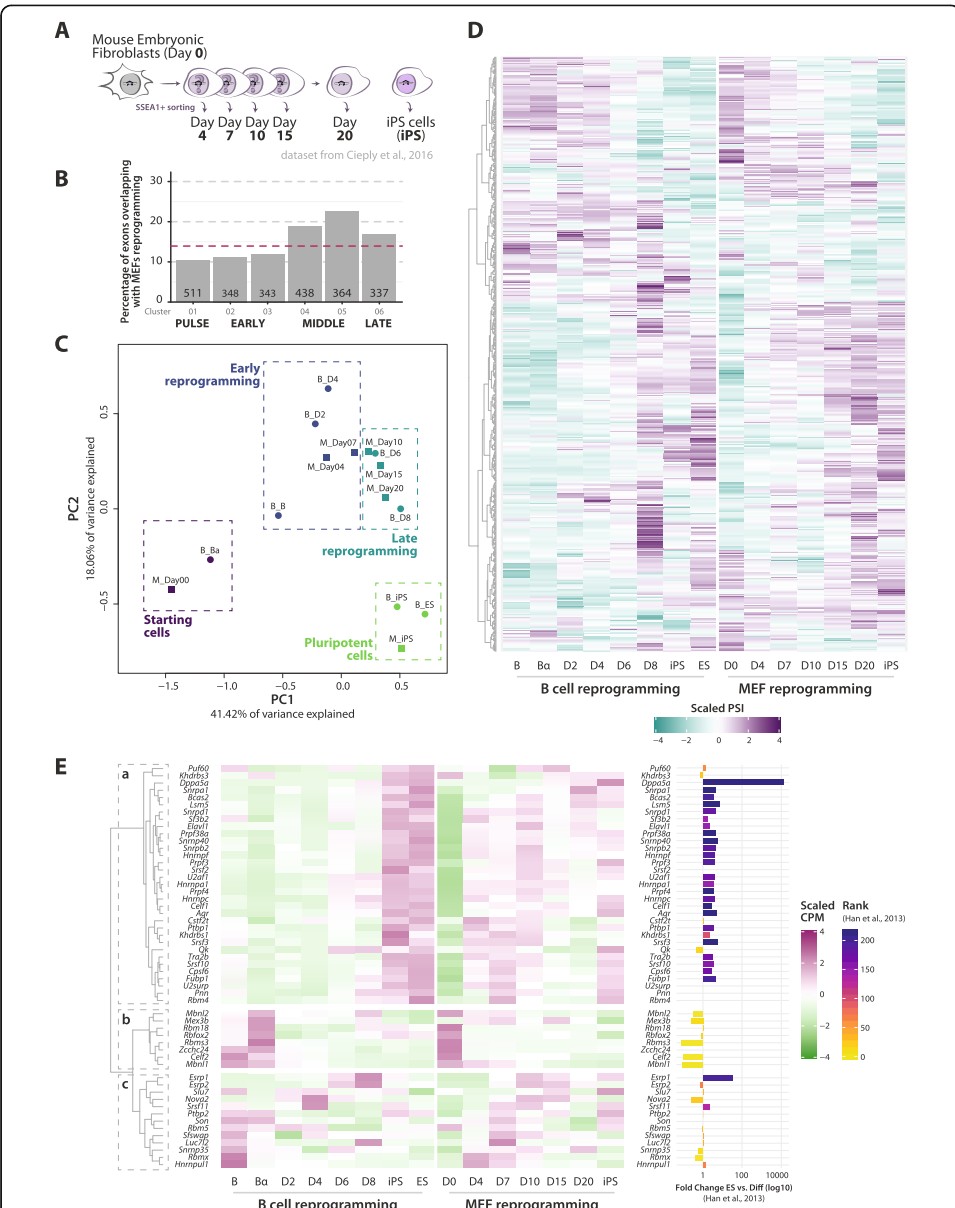

**Fig. 2** B cell and MEF reprogramming systems share a program of AS changes. **A** Schematic representation of MEF reprogramming time points analyzed by RNA-seq in [17]. **B** Percentage of exons in each B cell reprogramming AS cluster that are also detected as differentially spliced in the MEF reprogramming dataset of [17]. The number of events in each cluster is indicated at the bottom of the corresponding bar. The magenta dashed line indicates the average percentage for all 12 clusters. **C** PCA analysis of the 25% most variably expressed genes, segregated using *k*-means into 4 clusters: differentiated cells, early and late reprogramming and pluripotent cells, highlighted by colors and boxes. Circles: B cell reprogramming time points. Squares: MEF reprogramming time points. **D** Heatmap representing scaled PSI values (average between replicates) of exons differentially spliced in at least one time point in both B cell (left) and MEF reprogramming (right), with the corresponding hierarchical clustering. **E** Heatmap representing the expression of RNA-binding proteins known to be involved in pluripotency, somatic cell reprogramming, and/or development. Scaled cpm values (average between replicates) of both datasets are shown, with the corresponding hierarchical clustering. The bar plot (right) represents (when available) the fold change expression between ES cells and differentiated mouse tissues calculated in [7] and the corresponding ranking (color of the bar).

profiles, hierarchical clustering captured three known functional groups, with cluster (*a*) containing factors with higher expression in iPS/ES samples (fold change > 0 and high rank score according to 7, right panel) and known to promote pluripotency/reprogramming, such as *U2af1* or *Srsf2/3* [9, 14]. In contrast, cluster (*b*) contains factors with higher expression in the starting somatic cells, which includes known repressors of reprogramming such as *Mbnl1/2*, *Celf2*, and *Zcchc24* [7, 11, 17]. Finally, cluster (*c*) contains factors with more variegated expression patterns at early and intermediate reprogramming steps (and mildly repressed in iPS/ES cells), including *Esrp1/2* [16, 17] (Fig. 2E).

Taken together, our analyses revealed widespread AS changes during B cell reprogramming, which significantly overlapped with those of MEF reprogramming, especially at intermediate phases of the process.

### Predicted regulators of alternative splicing during somatic cell reprogramming

To infer potential regulators of exons differentially spliced during B cell reprogramming, we extracted gene expression profiles of 507 RBPs (as annotated in the *Uniprot* database), which were detectably expressed (cpm ≥ 5 in at least 33% of samples) and featured a minimum of variation across the B cell reprogramming dataset (coefficient of variation ≥ 0.2). Using the *membership* function of the *Mfuzz* package, we correlated (positively or negatively) the scaled gene expression profile of each RBP to each AS cluster centroid. This allowed us to derive a list of potential regulators whose changes in levels of expression correlate (or anti-correlate) with the profiles of AS changes in each cluster (membership > 0.3) (Fig. 3, Additional file 1: Figure S3 and Additional file 4: Table S3). In line with previous work [7, 11, 14], our analysis identified known AS regulators involved in the induction or repression of pluripotency/developmental decisions, such as *Mbnl1/2*, *Celf2* (both potential negative regulators of pulse/late clusters 1/6), and *U2af1* (potential positive regulator of middle cluster 4) (Fig. 3). Importantly, additional RBPs and splicing factors without previously known functions in reprogramming emerged as possible regulators.

To further define such regulators, we focused on RBPs that change their expression after the C/EBPα pulse or at early stages of reprogramming for functional validation during the induction of pluripotency, as we speculated that these could drive the inclusion/skipping of both early and intermediate AS exons during the reprogramming of C/EBPα-poised B cells. We selected CPSF3 as a putative positive regulator of very early events because its profile of expression increases during B cell and MEF reprogramming in parallel with centroid of AS changes in cluster 1 (Fig. 3A). While CPSF3 was originally described as part of cleavage / polyadenylation complexes [27], more recent work implicated components of this complex on alternative splicing regulation, including numerous internal exons not linked to the selection of alternative polyadenylation sites [28, 29]. Following a similar logic, we chose two putative negative regulators of cluster 1, namely TIA1, a well-described AS regulator with roles in cell proliferation and development (see below), and hnRNP UL1, a member of the heterogeneous ribonucleoprotein (hnRNP) family whose involvement in splicing regulation is largely unexplored (Fig. 3B). Due to experimental difficulties of genetic manipulation of the B cell reprogramming system (see "Discussion"), we modulated their expression at early

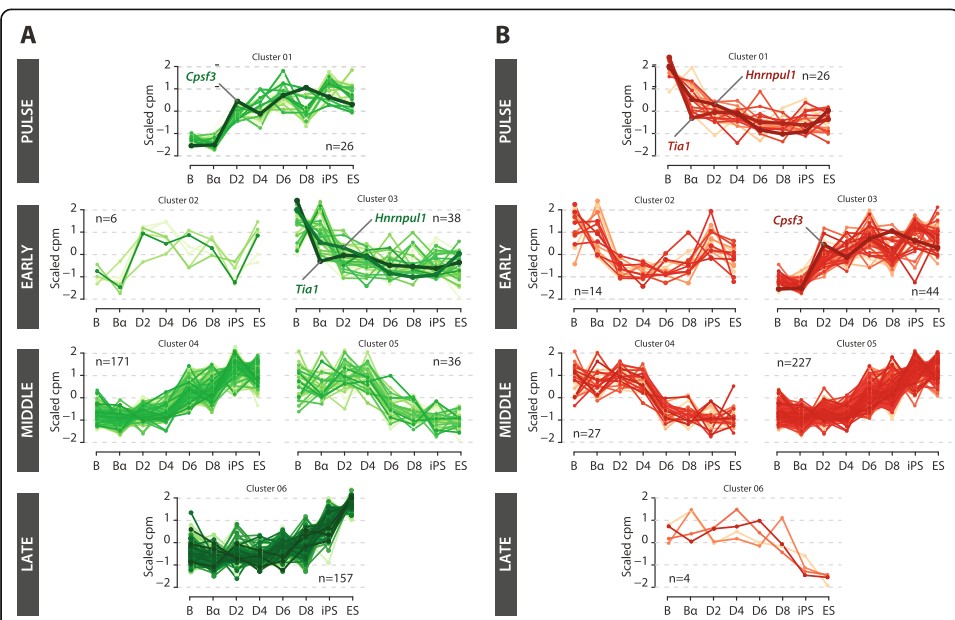

**Fig. 3** Inferred regulators of AS changes during B cell reprogramming. **A** Gene expression profiles of RNA-binding proteins (RBPs) correlating with the centroid of each AS cluster (positive regulators). Average scaled cpm values are represented by each line and the number of putative positive or negative regulators of each cluster is indicated (1). Expression profiles of *Cspf3*, *Hnrnpul1* and *Tia1* are highlighted. See also Additional file 1: Figure S3A. **B** Gene expression profiles of RBPs correlating with the negative of the centroid of each AS cluster (negative regulators). Displays as in panel **A**. See also Additional file 1: Figure S3B

stages of MEF reprogramming [30] and examined the consequences on the dynamics of pluripotency induction and on relevant AS alterations.

### Knockdowns of CPSF3 or hnRNP UL1 repress MEF reprogramming

The Cleavage and Polyadenylation Specificity Factor (CPSF) complex is primarily involved in mRNA polyadenylation, but a number of its components, including CPSF3, has been shown to also play a role in splicing [28, 29, 31–33]. *Cpsf3* expression increased early during reprogramming of both B cells and MEFs (Figs. 3A and 4A). The expression of *Hnrnpul1*, an hnRNP whose function in RNA metabolism is poorly understood, decreases in B cells after the C/EBPα pulse and then stabilizes at levels similar to those observed throughout MEF reprogramming (Figs. 3B and 4B).

To test their effects on MEF reprogramming, two different short hairpin RNAs (shRNAs) for each factor were cloned into lentiviral vectors containing a GFP reporter. The protocol used is summarized in Additional file 1: Figure S4A. Briefly, early passage MEFs isolated from reprogrammable mice were transduced with constructs bearing the shRNAs and the cells treated with doxycycline to activate OSKM (day 0). GFP+ cells were FACS-sorted 48 h post-infection and seeded on inactivated MEFs serving as feeder layers, to proceed with reprogramming for up to 14 days. Cells were harvested every other day and samples analyzed by RT-qPCR for the expression of the shRNAs target and of pluripotency markers, and by flow cytometry to quantify the proportion of cells expressing the early pluripotency cell surface marker SSEA1 (appearing around days 5–6) and the later pluripotency marker EPCAM1 (appearing around day 8, [34]). At day

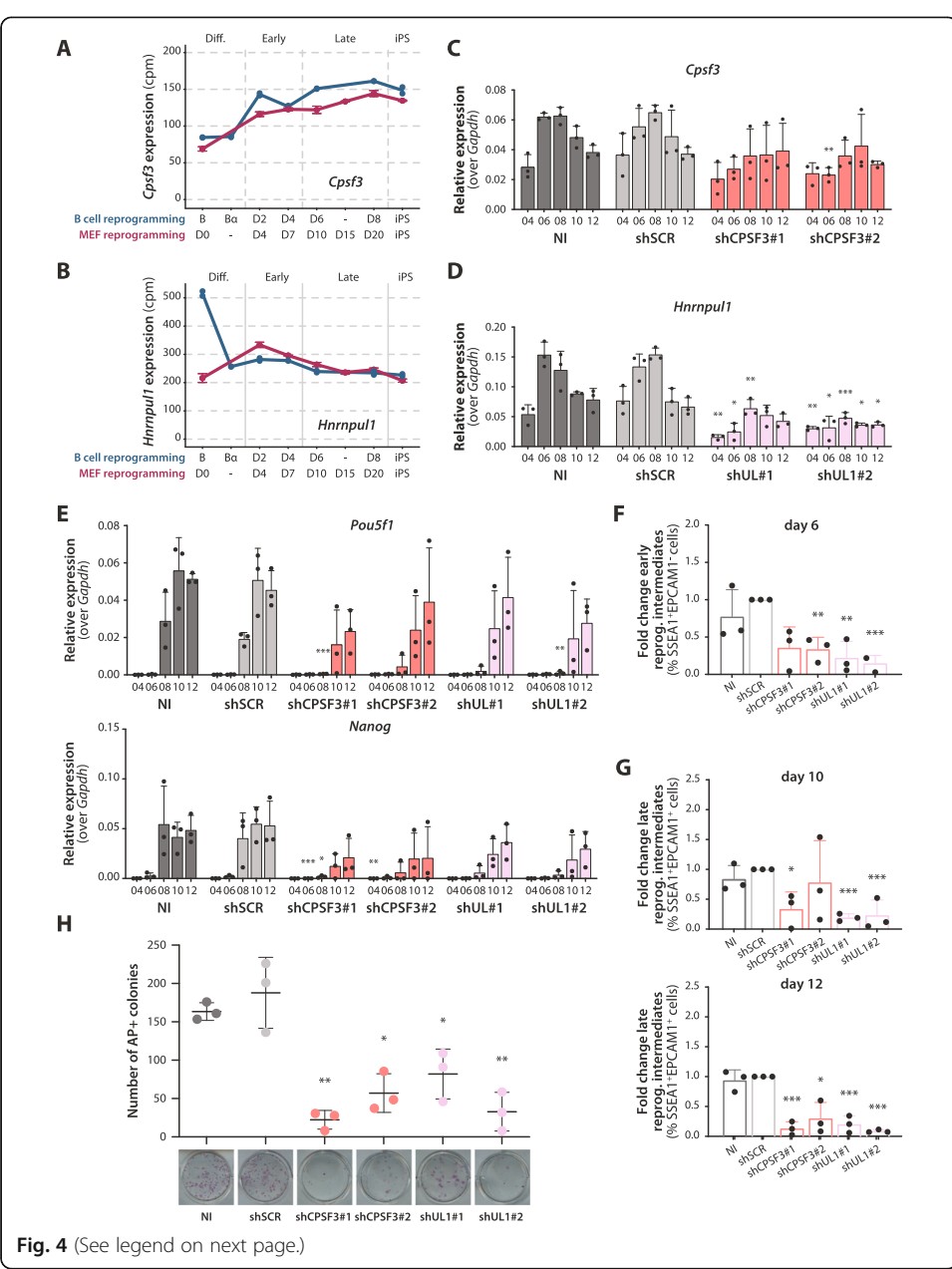

**Fig. 4** (See legend on next page.)

(See figure on previous page.)

**Fig. 4** Knockdowns of CPSF3 or hnRNP UL1 impair MEF reprogramming. **A** Gene expression profiles of *Cpsf3* in B cell reprogramming and MEF reprogramming (cpm values, blue and magenta lines, respectively). The *x*-axis represents "reprogramming pseudotime" in both datasets, calculated through the PCA analysis of Figure 2B. **B** Gene expression levels of *Hnrnpul1* in B cell and MEF reprogramming (cpm values, blue and magenta lines respectively), as in panel **A**. **C** Relative expression levels of *Cpsf3* mRNA quantified by RT-qPCR in non-infected cells (NI), cells transduced with a scrambled control shRNA (shSCR) or one of two shRNAs specific for *Cpsf3*. The *y*-axis represents the relative expression (2^(−ΔCt) value) of *Cpsf3* after normalization over *Gapdh*. **D** Relative expression levels of *Hnrnpul1* mRNA quantified by RT-qPCR as in panel **C**. **E** Relative expression levels of *Pou5f1* and *Nanog* quantified by RT-qPCR as in panels **C** and **D**. See also Additional file 1: Figure S4B. **B–E** Average of biological triplicates and SD values are shown. Statistical significance was calculated by *t*-test on ΔCt values compared to the NI control (*, **, *** = *p* value < 0.05, 0.01, 0.001 respectively, corrected for multiple testing using Holm-Sidak method). **F** Reduction of early reprogramming intermediates at day 6 post-OSKM induction upon knockdowns of *Cpsf3* and *Hnrnpul1*. Fold change was calculated from the percentage of SSEA1+EPCAM1− cells (of the total of alive cells) in every condition compared to the shSCR control using flow cytometry analysis. **G** Reduction of late reprogramming intermediates at days 10 and 12 post-OSKM induction upon knockdown of *Cpsf3* or *Hnrnpul1*. Fold change was calculated from the percentage of SSEA1+EPCAM1+ cells (of the total of alive cells) in every condition compared to the shSCR control using flow cytometry analysis. See Additional file 1: Figure S4C for examples of gates. **H** Number of colonies stained with alkaline phosphatase (AP) at day 14 post-OSKM induction upon knockdown of *Cpsf3* or *Hnrnpul1*. On the bottom, images of representative wells are shown for every condition. **F,G,H** Average of biological triplicates and SD values are shown. Statistical significance was calculated by *t*-test comparing each condition to the shSCR control (*, **, *** = *p* value < 0.05, 0.01, 0.001 respectively, corrected for multiple testing with Holm-Sidak method)

14, 2 days after removing doxycycline, cultures were stained for alkaline phosphatase (AP) activity to identify iPS colonies and to assess the efficiency of reprogramming.

Cells infected with shRNAs targeting *Cpsf3* and *Hnrnpul1* were compared with non-infected (NI) cells, as well as with cells transduced with a scrambled control shRNA (shSCR). Knockdown efficiency, quantified by RT-qPCR (Fig. 4C, D), showed a 51% and 58% reduction in *Cpsf3* at day 6 post-infection with shCPSF3#1 and #2, respectively, becoming slightly less efficient during reprogramming (Fig. 4C). Similarly, knockdown of *Hnrnpul1* resulted in 81% and 76% reduction of mRNA levels at day 6 post-infection with shUL1#1 and #2, respectively (Fig. 4D). The increase in the mRNA levels of endogenous *Pou5f1* (encoding OCT4) and *Nanog* pluripotency markers was delayed in both *Cpsf3* and *Hnrnpul1* knockdown cells (compare for example values at day 8 in Fig. 4E and Additional file 1: Figure S4B), suggesting slower reprogramming kinetics. Consistent with these observations, knockdown of *Cpsf3* and *Hnrnpul1* reduced the percentage of cells expressing SSEA1 at day 6 (Fig. 4F) and cells expressing both SSEA1 and EPCAM1 at days 10–12 (Fig. 4G and Additional file 1: Figure S4C). Survival of cells during reprogramming did not seem to be affected in the knockdowns because no significant increase in the proportion of DAPI-stained cells was observed throughout reprogramming (Additional file 1: Figure S4D). Finally, we counted the number of AP positive colonies at day 14 and found that the amount was significantly reduced in cells infected with either the shRNAs targeting *Cpsf3* or those targeting *Hnrnpul1* (Fig. 4H). We finally tested the effects of overexpression of a T7-tagged version of CPSF3 from the beginning of reprogramming (Additional file 1: Figure S4A and E). A trend towards increased expression of *Pou5f1* and *Nanog* at late times of reprogramming upon *T7-Cpsf3* overexpression was observed (Additional file 1: Figure S4F). However, this was not accompanied by enhanced reprogramming efficiency or redistribution of reprogramming intermediates (Additional file 1: Figure S4G-I), likely because of additional

rate-limiting steps required to increase the efficiency of this complex process or because of suboptimal timing or levels of overexpression achieved in the experiment.

Taken together, these data show that the knockdown of *Cpsf3* or *Hnrnpul1* reduces the MEF to iPS reprogramming efficiency, therefore arguing that both RBPs contribute to the induction of pluripotency.

### Overexpression of TIA1 represses MEF reprogramming

TIA1 is an RBP and AS regulator [35–37]. *Tia1* mRNA levels decrease early during B cell reprogramming (Figs. 3B and 5A), compatible with a potential role as a repressor of cell reprogramming in this system. Because depletion of TIA1 induces mouse embryonic lethality and the protein is important for MEF proliferation, cell cycle progression, autophagy, and numerous signaling pathways [38], we decided to test the effects of TIA1 overexpression during MEF reprogramming. For this purpose, primary MEFs were infected at day 0 (concomitantly with the induction of reprogramming), with retroviral constructs containing a T7-tagged *Tia1* cDNA and a GFP reporter to allow sorting of the transduced cells (Additional file 1: Figure S4A). Levels of *Tia1* were quantified by RT-qPCR (24-fold increase compared to cells transduced with an empty vector, Fig. 5B). Consistent with our prediction from its expression profile in B cell reprogramming, overexpression of *Tia1* repressed the induction of endogenous *Pou5f1* and *Nanog* genes (Fig. 5C and Additional file 1: Figure S5A) and led to a reduction of early SSEA1$^+$EPCAM1$^-$ cells and of late reprogramming intermediates (SSEA1$^+$EPCAM1$^+$ cells) compared to empty vector and non-infected controls (Fig. 5D, E and Additional file 1: Figure S5B). In addition, it significantly reduced the count of AP$^+$ colonies at day 14 post-OSKM induction compared to controls (Fig. 5F), without substantially affecting the viability of reprogramming cells (Additional file 1: Figure S5C). Of note, overexpression of *Tia1* delayed the gradual skipping of *Lef1* exon 6, a conserved AS event between B cell and MEF reprogramming (Fig. 5G), which might partially explain the observed reduction in reprogramming efficiency.

Taken together, the observed reduced expression of TIA1 during B cell reprogramming and its impairment of MEF reprogramming when overexpressed suggest that it functions as a general repressor of pluripotency induction.

### CPSF3, hnRNP UL1, and TIA1 regulate alternative splicing during reprogramming

To assess the effects of CPSF3, hnRNP UL1, and TIA1 manipulation on AS during reprogramming, RNA-seq analyses were carried out with RNAs isolated from MEFs at 0 and at 12 days post-OSKM induction, comparing the effects of *Cpsf3/Hnrnpul1* knockdown (two different shRNAs for each factor) or *Tia1* overexpression with those of the corresponding shSCR/empty vector controls (Fig. 6A). Quantification of gene expression of *Tia1*, *Cpsf3*, and *Hnrnpul1* at the two timepoints is shown in Additional file 1: Figures S6A-B.

To determine the impact of these perturbations on AS, we calculated differentially spliced events between day 0 and day 12 in each condition as described before (|ΔPSI(-day12 – day0)| ≥ 10, range ≥ 5). TIA1-dependent events were defined as those changing during reprogramming only in the control or the overexpression condition, with |ΔΔPSI(TIA1 – Empty)| ≥ 10 (colored dots in Fig. 6B). Similarly, CPSF3-dependent

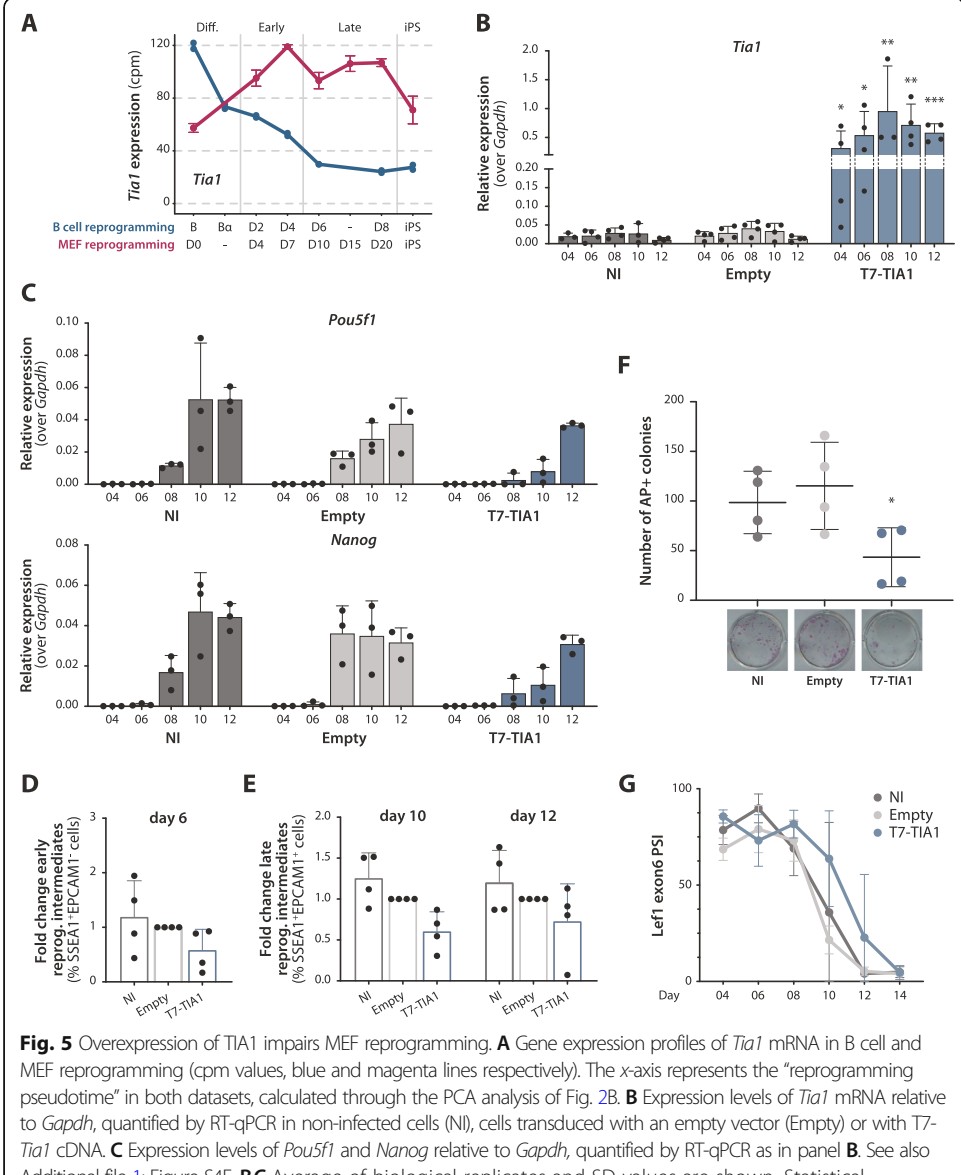

**Fig. 5** Overexpression of TIA1 impairs MEF reprogramming. **A** Gene expression profiles of *Tia1* mRNA in B cell and MEF reprogramming (cpm values, blue and magenta lines respectively). The *x*-axis represents the "reprogramming pseudotime" in both datasets, calculated through the PCA analysis of Fig. 2B. **B** Expression levels of *Tia1* mRNA relative to *Gapdh*, quantified by RT-qPCR in non-infected cells (NI), cells transduced with an empty vector (Empty) or with T7-*Tia1* cDNA. **C** Expression levels of *Pou5f1* and *Nanog* relative to *Gapdh,* quantified by RT-qPCR as in panel **B**. See also Additional file 1: Figure S4E. **B,C** Average of biological replicates and SD values are shown. Statistical significance was calculated by *t*-test on ΔCt values comparing to the Empty control (\*, \*\*, \*\*\* = *p* value < 0.05, 0.01, 0.001 respectively, corrected for multiple testing with Holm-Sidak method). **D** Percentage of SSEA1+EPCAM1− early reprogramming intermediates (day 6 post-OSKM induction) upon *Tia1* overexpression determined by flow cytometry. **E** Percentage of SSEA1+EPCAM1+ late reprogramming intermediates (days 10 and 12 post-OSKM induction) upon *Tia1* overexpression. See Additional file 1: Figure S5B for gating strategy. **F** Number of alkaline phosphatase (AP) positive colonies at day 14 post-OSKM induction upon *Tia1* overexpression. Images of representative wells are shown below. **D,E,F** Average of biological replicates and SD values (*n* = 4). Statistical significance was calculated by *t*-test comparing to the Empty control (\*, \*\*, \*\*\* = *p* value < 0.05, 0.01, 0.001 respectively, corrected for multiple testing with Holm-Sidak method). **G** Inclusion of *Lef1* exon 6 upon overexpression of T7-*Tia1*, quantified by semi-quantitative RT-PCR and capillary electrophoresis. Values represent average and SD. Statistical significance calculated by *t*-test on average ± SD of the area under the curve in each condition yielded a *p* value of 0.079

and UL1-dependent events were defined as those with |ΔΔPSI(average shRNAs − shSCR)| ≥ 10 (only events selected by both specific shRNAs were considered, 54% and 60% overlap respectively, colored dots in Fig. 6C and Additional file 1: Figure S6C). For

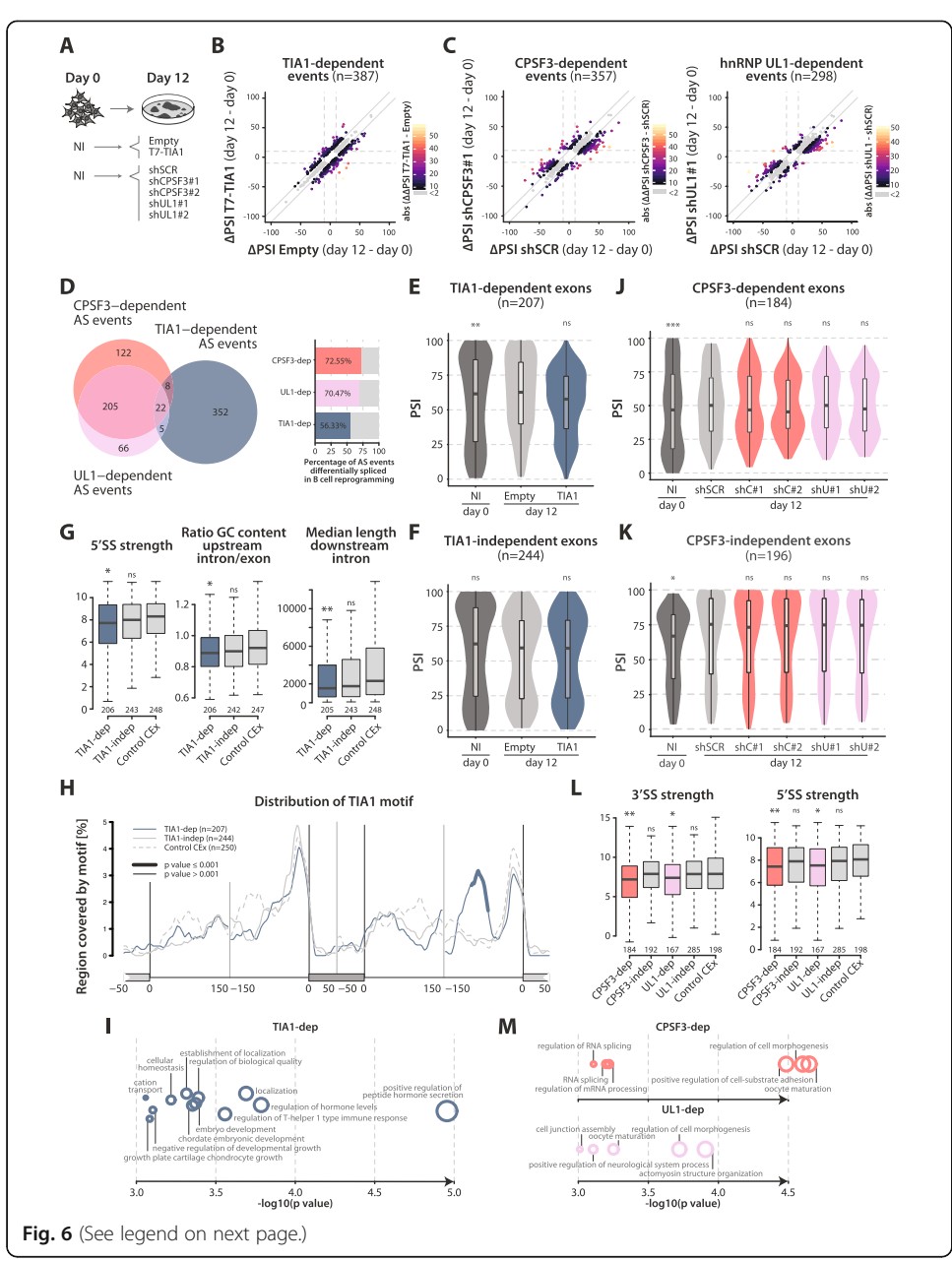

**Fig. 6** (See legend on next page.)

(See figure on previous page.)

**Fig. 6** Knockdowns of CPSF3 or hnRNP UL1 and overexpression of TIA1 regulate AS during reprogramming. **A** Schematic representation of the experiments performed to study the effect of TIA1 overexpression / CPSF3 or hnRNP UL1 knockdown by RNA-seq. **B** TIA1-dependent events detected during reprogramming. The *x*-axis represents the ΔPSI value between Empty day 12 and day 0. The *y*-axis represents the ΔPSI value between T7-TIA1 day 12 and day 0 control. TIA1-dependent events (|ΔΔPSI(T7-TIA1 − Empty)| ≥ 10) are represented by colored dots (palette representing the |ΔΔPSI(T7-TIA1 − Empty)| value, *n* = 387). TIA1-independent events (|ΔΔPSI(T7-TIA1 − Empty)| < 2) are represented by grey dots (*n* = 558). **C** CPSF3- and UL1-dependent events detected during reprogramming (left and right, respectively). The *x*-axis represents the ΔPSI value between shSCR day 12 and day 0 control. The *y*-axis represents the ΔPSI value between shCPSF3#1 or shUL1#1 day 12 and day 0 control. CPSF3/UL1-dependent events (ΔΔPSI(average_shRNAs − shSCR) ≥ 10) are represented by non-grey-colored dots (palette representing the ΔΔPSI(average_shRNAs − shSCR) value). CPSF3/UL1-independent events (ΔΔPSI(average_shRNAs − shSCR) < 2) are represented by grey dots. See Additional file 1: Figure S6C for ΔPSI values of the same events in shRNA#2 conditions. **D** Venn diagram representing the overlap between CPSF3-, UL1-, and TIA1-dependent events (left, the number of events in each category is shown). Barplot representing the percentage of CPSF3-, UL1-, and TIA1-dependent events which are also differentially spliced in B cell reprogramming (right, the percentage of overlap in each category is indicated). **E** Violin plots representing the distribution of PSI values of TIA1-dependent events in non-infected cells (NI, day 0) and day 12 cells infected with Empty or *T7-Tia1* vectors. **F** Violin plots representing the distribution of PSI values of TIA1-independent events as in panel **E**. **E,F** Statistical significance was calculated by Fisher's exact test comparing number of events with intermediate (25 < PSI < 75) or extreme PSI values (PSI ≥ 75 or ≤ 25) in each condition against Empty control (*, **, *** = *p* value < 0.05, 0.01, 0.001 respectively). See also Additional file 1: Figure S6D. **G** Boxplots representing the distribution of the indicated sequence features of TIA1-dependent exons, compared to TIA1-independent events and a random set of exons with intermediate PSI values not changing throughout reprogramming (Control CEx). Statistical significance was calculated using *Matt*, by paired Mann-Whitney *U* test comparing each condition to the Control CEx set (*, **, *** = *p* value < 0.05, 0.01, 0.001 respectively). **H** RNA map representing the distribution of TIA1 binding motif in TIA1-dependent exons and flanking introns, compared to TIA1-independent and Control CEx. Thicker segments indicate regions in which enrichment of TIA1 motif is significantly different compared to Control CEx. **I** Gene Ontology (GO) terms enriched in genes containing TIA1-dependent events, compared to a background of all genes containing mapped AS events in the dataset. GO enrichment was calculated using *GOrilla* and GO terms were summarized for visualization using *REViGO*. The *x*-axis and the size of each bubble represent the −log10(*p* value) of each GO term. **J** Violin plots representing the distribution of PSI values of CPSF3-dependent events in non-infected cells (NI, day 0) and day 12 cells infected with shSCR or two shRNAs specific for CPSF3 (shC#1 and shC#2) or UL1 (shU#1 and shU#2). **K** Violin plots representing the distribution of PSI values of CPSF3-independent events as in panel **J**. **J,K** Statistical significance was calculated by Fisher's exact test comparing number of events with intermediate (25 < PSI < 75) or extreme PSI values (PSI ≥ 75 or ≤ 25) in each condition against shSCR control (*, **, *** = *p* value < 0.05, 0.01, 0.001 respectively). See also Additional file 1: Figure S6E. **L** Boxplots representing the distribution of sequence features of CPSF3- and UL1-dependent exons, compared to the corresponding CPSF3- and UL1-independent events and Control CEx. Statistical significance was calculated using *Matt*, by paired Mann-Whitney *U* test comparing each condition to the Control CEx set (*, **, *** = *p* value < 0.05, 0.01, 0.001 respectively). **M** GO terms enriched in genes containing CPSF3- and UL1-dependent events, compared to a background of all genes containing mapped AS events in the dataset, performed as in panel **I**

each RBP, control events, changing during reprogramming but not affected by TIA1, CPSF3, or hnRNP UL1 manipulation, were classified as those differentially spliced between days 0 and 12, either in control or knockdown/overexpression conditions (|ΔPSI(day12 − day0)| ≥ 10, range ≥ 5), that display minimal differences between the ΔPSI values in the two conditions (e.g., |ΔΔPSI(TIA1 − Empty)| < 2) (grey dots in Fig. 6B, C and Additional file 1: Figure S6C). All sets are summarized in Additional file 5: Table S4.

We thus identified 387 TIA1-dependent events and 558 TIA1-independent events. Similarly, we identified 357 CPSF3-dependent and 298 UL1-dependent events (as well as 429 CPSF3-independent and 662 UL1-independent events). Interestingly, CPSF3-dependent and UL1-dependent events showed a significant overlap (e.g., 70% of UL1-dependent events were also CPSF3-dependent events, of which 98% changed in the same

direction) whereas little overlap was observed with TIA1-dependent events (Fig. 6D, left panel). As we found no strong evidence for cross-regulation between CPSF3 and hnRNP UL1 factors (either in AS or in gene expression, Additional file 1: Figure S6B), these data suggest that CPSF3 and hnRNP UL1 contribute to a common program of AS changes relevant for cell reprogramming. Moreover, TIA1-, CPSF3-, and UL1-dependent events detected in MEFs showed high overlap with events that were also differentially spliced during B cell reprogramming, suggesting that these factors regulate AS events that generally contribute to the induction of pluripotency (Fig. 6D, right panel).

Focusing on cassette exon events, TIA1-dependent exons (but not TIA1-independent exons) displayed significantly higher PSI values at day 12 compared to day 0 of reprogramming, as well as a relative increase in exons displaying intermediate PSI values (Fig. 6E, F and Additional file 1: Figure S6E). These effects are altered by TIA1 overexpression (Fig. 6E), suggesting that in this system, the typical role of TIA1 is to repress exon inclusion.

Analysis of sequence features associated with TIA1-regulated exons performed using *Matt* [39] revealed weaker 5′ splice sites, shorter median length of the flanking downstream introns and a larger difference in GC content between the alternative exons and their flanking upstream introns (Fig. 6G), suggestive of a strong dependence of these exons on the process of exon definition [40]. Notably, an enrichment in putative TIA1 binding motifs was detected about 100 nucleotides upstream of the distal 3′ splice site (Fig. 6H), suggesting that binding of TIA1 to this region might repress inclusion of the alternative exon (or enhance pairing between the distal splice sites), leading to exon skipping.

Gene Ontology (GO) enrichment analysis (*GOrilla*; [41]) on the set of genes containing TIA1-dependent AS events compared to a background list of all genes containing mapped AS events ($n$ = 11132) showed an enrichment in functions related to peptide/hormone secretion, T-helper cell functions, and embryonic development (Fig. 6I and Additional file 6: Table S5).

CPSF3- and UL1-dependent exons showed similar inclusion patterns: an increased proportion of intermediate PSIs was observed at day 12 in CPSF3- and UL1-dependent but not in CPSF3- and UL1-independent events (Fig. 6J, K and Additional file 1: Figure S6E). CPSF3- and UL1-dependent exons also showed higher PSI values at day 12 compared to day 0, and an enrichment in exons displaying intermediate PSI values at the end of the reprogramming process (Fig. 6J, K and Additional file 1: Figure S6E). These effects were attenuated, specifically for CPSF3- and UL1-dependent exons, upon knockdown of either of these factors (Fig. 6J, K and Additional file 1: Figure S6E), supporting the notion that their regulatory programs overlap. Both CPSF3- and UL1-dependent exons displayed weaker 3′ and 5′ splice sites (Fig. 6L). GO analyses revealed that CPSF3- and UL1-dependent events belong to genes enriched in functional categories related to the regulation of cell morphogenesis, cell-substrate adhesion, and cytoskeleton organization (Fig. 6M and Additional file 6: Table S5). Changes in inclusion levels of selected CPSF3- and UL1-dependent events were validated by semi-quantitative RT-PCR (Additional file 1: Figure S6F).

As CPSF3 was originally described as a cleavage / polyadenylation factor, we asked whether the CPSF3-regulated exons could be the result of changes in alternative polyadenylation (APA) sites. Not unexpectedly, quantification of poly-A usage performed

with QAPA [42] revealed hundreds of genes with APA events affected by *Cpsf3* knockdown during reprogramming (277 genes) and, interestingly, also many genes with APA events affected by *Hnrnpul1* knockdown (237 genes) ($|\Delta\Delta PAU(\text{shRNA}-\text{shSCR}) \geq 10$). However, the overlap between the genes / exons whose AS or APA were affected by *Cpsf3* or *Hnrnpul1* knockdown was very limited (Additional file 1: Figures S6G-H), suggesting that the mechanisms by which CPSF3 and hnRNP UL1 regulate alternative splicing and 3′ end processing are distinct and independent.

Taken together, these results suggest that the AS regulators TIA1, CPSF3, and hnRNP UL1 function in cell reprogramming through their activities on genes relevant for cell fate decisions. Moreover, CPSF3 and hnRNP UL1 act through a highly overlapping program of splicing changes, while TIA1 affects a different set of AS events. Distinct programs of AS and APA were associated with CPSF3 and hnRNP UL1.

## Discussion

Somatic cell reprogramming holds great promise for fundamental research and medicine. In this study, we have characterized the dynamics of AS changes during the deterministic reprogramming of B cells into iPS cells and have identified three splicing regulators with a role in somatic cell reprogramming, as validated in perturbation experiments performed during MEF reprogramming.

The B cell reprogramming system is a two-step process, in which a pulse of the myeloid-specific transcription factor C/EBPα poises the cells for deterministic reprogramming by the subsequent induction of the OSKM factors, inducing early chromatin opening and epigenetic modifications that silence of the B cell-specific program and induce pluripotency [18–20, 43]. While C/EBPα might not directly impact AS, it remains possible that the pulse of C/EBPα could induce changes in the expression of splicing regulatory factors (e.g., induction of CPSF3 or repression of hnRNP UL1 or TIA1) that trigger a program of post-transcriptional changes that contributes to cell reprogramming or that C/EBPα could have direct effects on the activity of splicing factors.

We observed widespread changes in AS, whose frequency increased as reprogramming progressed. Clustering analysis of cassette exons revealed groups of exons varying in inclusion at very early stages of B cell reprogramming (including following the C/EBPα pulse), but also at middle and late phases after OSKM induction. Of note, early clusters contain a larger proportion of exons predicted to disrupt open reading frames (ORFs) of isoforms predominantly expressed in the B cells, whereas middle clusters contain more ORF-preserving exons. Together with the observation that the majority of regulated introns tend to be retained (leading to ORF disruption) at early stages and are progressively spliced during reprogramming, these findings suggest that, in the transition from early to late stages, AS acts as a switch towards a gain in coding capacity, while at intermediate stages of reprogramming AS acts more commonly as a switch between isoforms.

The AS switch observed between day 4 and day 6 of B cell reprogramming coincides with a transcriptional switch, although the proportion of genes displaying both gene expression and AS changes is lower than expected (typically 1/3 of alternatively spliced transcripts are controlled at the level of gene expression by nonsense-mediated decay [1, 26, 27, 44]. The transcriptomic changes observed during this transition might be mediated at least in part by the replacement of the culture medium from serum + LIF cell to N2B27-based 2i medium

at day 6. Exons changing their inclusion levels during this of intermediate period displayed the highest overlap with exons differentially spliced in the reprogramming of MEFs, arguing for the general relevance of these changes in the induction of pluripotency.

Our analysis of changes in expression of RBPs during B cell reprogramming suggested potential regulators of AS, including factors already known to play a role in cell reprogramming such as the pluripotency repressors MBNL1/2 and CELF2 [7, 11], which are downregulated during B cell reprogramming, and positive pluripotency regulators such as U2AF1 and hnRNP H1 [10, 14], which were upregulated during the process, consistent with results described for MEF reprogramming [17]. We focused on three candidate regulators of early AS changes, CPSF3, hnRNP UL1, and TIA1, because they could potentially contribute to the reprogramming advantage acquired by B cells poised by C/EBPα and/or to early events occurring during reprogramming. We found that perturbations of each of the three factors impaired the reprogramming of MEFs. We focused on functional tests in MEFs because of the more technical challenging two-step protocol required for reprogramming B cells, the lower cellular yields, and the lower efficiency of transfection/infection of B cells, which greatly complicates knockdown and overexpression experiments. CPSF3 plays an important role in mRNA polyadenylation, recognizing the polyadenylation signal (PAS) together with the rest of the CPSF complex. In addition to the well-known intense crosstalk between last intron splicing and 3′ end processing [31], CPSF2 and the CPSF complex have been shown to influence splicing of internal exons independently of cleavage and polyadenylation [28, 29, 32]. Furthermore, mutations of the CPSF3 yeast homolog Brr5/Ysh1 have been shown to strongly affect splicing, suggesting that CPSF3 could also play similar direct regulatory functions [33]. hnRNP UL1, instead, is a largely uncharacterized RBP known to participate in DNA damage response [45] and identified as a surface marker of human ES cells, although its functions in this context remain unknown [46]. Recent work in Zebrafish suggests a possible role for Hnrnpul1 in the regulation of AS [47].

Knockdown of *Cpsf3* or *Hnrnpul1* (with two independent shRNAs for each gene) caused a general repression or delay in MEF reprogramming, revealing their functions as promoters of somatic cell reprogramming. Sequence analysis of regulated AS events suggest that CPSF3 and hnRNP UL1 favor the definition of exons harboring relatively weak splice sites. Remarkably, the AS targets of CPSF3 and hnRNP UL1 identified by our RNA-seq analyses show a high overlap (without evidence of cross-regulation between the two proteins), suggesting a concerted mechanism of splice site selection, perhaps as components of the same complex. It is unlikely that the overlap is an indirect consequence of reduced/delayed reprogramming, because TIA1 overexpression also inhibits reprogramming and yet it is accompanied by largely non-overlapping changes in AS. Our results therefore suggest that multiple, separable AS programs contribute to the regulation of cell reprogramming and that these factors can contribute to reprogramming based upon their activities as splicing regulators, alternative polyadenylation regulators, or both.

TIA1 is a regulator of RNA metabolism influencing mRNA decay and AS implicated in cell proliferation, cell cycle progression, cellular stress, autophagy, and programmed cell death [35, 37, 38, 48] whose depletion induces mouse embryonic lethality [49].

We found that TIA1 overexpression at early stages of reprogramming represses the induction of pluripotency, as reported for MBNL proteins [7]. Binding of TIA1 to U-

rich sequences downstream of weak 5′SS helps to recruit U1 snRNP and facilitates exon definition [35–37, 50], while it can also inhibit the inclusion of alternative exons located at a distance from its binding site [51]. As expected, TIA1-dependent exons are enriched in weak 5′ splice sites and other features that support a role in exon definition. Interestingly, TIA1-dependent exons in our system feature an enrichment of predicted U-rich TIA1 binding sites approximately 100 nucleotides upstream of the 3′ splice site located in the downstream intron, suggesting that this configuration serves to repress the inclusion of exons that contribute to cell reprogramming. Since GO terms most enriched in genes bearing TIA1-dependent exons include "positive regulation of peptide hormone secretion," we hypothesize that TIA1-regulated transcripts could modulate functions related to the senescence-associated secretory phenotype (SASP), which was shown to promote reprogramming of somatic cells and dedifferentiation in cancer [52–54]. Indeed, 7 out of 23 (30%) of the genes belonging to these GO terms were also found in a list of senescence- or SASP-associated genes (*GeneCards* database; [55]).

The finding that overexpression of TIA1 caused a delay in the conserved gradual skipping of *Lef1* exon 6 observed during MEF reprogramming is consistent with a conserved role of the factor in reprogramming, as gradual skipping of this exon was also observed in middle stages of B cell reprogramming along with a decrease in expression of the gene. LEF1 is a transcription factor with important functions in embryonic and T cell development and activation [56–60]. As *Lef1* exon 6 inclusion is promoted by CELF2 during T cell development and activation [61–63] and *Celf2* expression rapidly decreases in both B cell and MEF reprogramming, CELF2 may participate—along with TIA1—in *Lef1* exon 6 regulation during reprogramming. We observed that overexpression of *Lef1* at the start of MEF reprogramming improved reprogramming efficiency (observed as an increase of pluripotency markers and percentage of reprogramming intermediates), with overexpression of the iPS-associated *Lef1* exon 6-skipping isoform displaying stronger effects than the exon 6-including isoform, suggesting a functional role for this switch in cell reprogramming (Additional file 1: Figures S5D-K). We speculate that the gradual skipping of exon 6 affects the interaction of LEF1 with protein partners that either act as coactivators—e.g., ALY [64]—or corepressors—e.g., Groucho/TLE [65], differentially affecting the transcription of target genes. For example, the LEF1-inclusion isoform could promote cell proliferation and maintenance of B cell identity—as observed in the C/EBPα-dependent transdifferentiation system and in pancreatic cancer cell lines [66, 67], while gradual skipping of exon 6 might alter its effect on its target genes with the consequence of inducing pluripotency, most probably in a β-catenin-independent way.

## Conclusions

Our results provide a comparison between AS changes occurring in two very different reprogramming systems and a wealth of information relevant for cell fate decisions and transitions to pluripotency. Furthermore, they demonstrate the functional involvement of the splicing regulators CPSF3, hnRNP UL1, and TIA1 as activators or repressors of efficient cell reprogramming. Our work significantly extends previous knowledge on RNA processing during reprogramming and suggests that AS changes play functional

roles during the entire transition of somatic into pluripotent stem cells. It would be interesting to determine whether the AS/APA changes and regulators identified in our work play similar roles during the specification of pluripotent stem cells in early embryo development.

## Methods

### RNA sequencing

For RNA-seq of B cell reprogramming and MEF reprogramming upon TIA1 overexpression and knockdown of CPSF3 and hnRNP UL1, stranded libraries were prepared from samples and sequenced on an Illumina HiSeq 2500 using a 2x125nt paired-end protocol (see "Availability of data and materials"). Duplicates were sequenced for each condition, with samples pooled in separate lanes. RNA-seq data (triplicates of 2x100nt paired-end sequencing) of MEF reprogramming was downloaded from GEO Database ([68], see "Availability of data and materials").

### Gene expression and alternative splicing analyses

Reads mapping (mm10 annotation) was performed with STAR v2.7.1a [69] and gene expression analysis was performed using the *edgeR* package v3.16.5 [25, 70]. A minimum of 5 counts per million (cpm) was required in 33% of both datasets (5 samples for B cell reprogramming, 6 samples for MEF reprogramming) and a minimum coefficient of variation of 0.2 was required.

Alternative splicing (AS) analysis was performed using *vast-tools* software v2.2.2 [21, 71]. Strand-specific mapping was performed and only events with a minimum of 10 actual reads per sample were considered (VLOW quality score). PSI values for single replicates were quantified for all types of alternative events, including simple and complex cassette exons (S, C1, C2, C3), microexons (MIC), alternative 5′ and 3′ splice sites (Alt5, Alt3) and retained introns (IR-S, IR-C). For cassette exon events (referred to as CEx), PSI values of all annotated exons were also quantified with *vast-tools* (Annotation module ANN). For both B cell and MEF reprogramming, all possible pairwise comparisons between samples were performed, selecting differentially spliced events with a $|\Delta PSI| \geq 10$ and a minimum range of 5 between samples.

For TIA1 overexpression experiments, TIA1-dependent events were defined as events with $|\Delta\Delta PSI| \geq 10$, where $\Delta\Delta PSI = (\Delta PSI \text{ TIA1\_day12} - \text{NI\_day0}) - (\Delta PSI \text{ Empty\_day12} - \text{NI\_day0})$. TIA1-independent events were instead defined as events changing in any of the two conditions $(|\Delta PSI| \text{ TIA1\_day12} - \text{NI\_day0} \geq 10)$ or $(|\Delta PSI| \text{ Empty\_day12} - \text{NI\_day0} \geq 10)$ and having minimal difference between the two conditions $(|\Delta\Delta PSI| < 2)$.

Similarly, CPSF3- or UL1-dependent events were events for which in both shRNAs $|\Delta\Delta PSI| \geq 10$, where $\Delta\Delta PSI = (\Delta PSI \text{ average\_shC/U\_day12} - \text{NI\_day0}) - (\Delta PSI \text{ shSCR\_day12} - \text{NI\_day0})$. CPSF3- or UL1-independent events were instead defined as events changing in any of the knockdown or control conditions $(|\Delta PSI| \text{ average\_shC/U\_day12} - \text{NI\_day0})$ or $(|\Delta PSI| \text{ shSCR\_day12} - \text{NI\_day0} \geq 10)$ and having minimal difference between the two conditions $(|\Delta\Delta PSI| < 2)$.

### Clustering analysis of alternatively spliced exons and correlation of gene expression profiles

PSI values of exons differentially spliced in at least one pair of B cell reprogramming samples ($n$ = 4556) were scaled and centered (referred to as "scaled PSI" values). Fuzzy $c$-means clustering analysis was carried out using the R package *Mfuzz* [22, 72] on the scaled PSI values. The optimal number of clusters (12) was selected on the basis of the minimum distance to the cluster centroid. A *membership* value was assigned to correlate gene expression profiles to each AS clusters, either of genes containing the AS exons of each cluster or of genes encoding RBPs. To each scaled cpm values vector, we attributed a *membership* value to the centroid of each cluster or to its negative.

### Correlation of B cell reprogramming stages

Pearson correlation coefficient for B cell reprogramming stages was calculated on the most variable expressed genes (cpm ≥ 5 in at least 5 samples and coefficient of variation ≥ 0.73864, corresponding to the 3rd quartile, $n$ = 2961) or the most variable exons (differentially spliced in at least one pairwise comparison and coefficient of variation ≥ 0.4107, corresponding to the 3rd quartile, $n$ = 1139).

### Prediction of the protein impact of alternative exons

Alternative exons detected by *vast-tools* were classified as described in *vastDB* v2.2.2 [21]. Briefly, following division according to their location in non-coding RNAs, untranslated regions (5′ or 3′ UTRs) or open reading frame (ORF), exons were predicted to disrupt the ORF if their inclusion or skipping would induce a frameshift in their ORF or if they would induce a premature stop codon (PTC) predicted to be targeted by nonsense-mediated decay (NMD) or truncating the protein by more than 300 amino acids. The rest of the ORF-mapping events were predicted to preserve the transcript coding potential.

### Principal component analysis (PCA) and comparison between B cell and MEF reprogramming

Gene expression values were filtered for minimum variation (coefficient of variation ≥ 0.66, corresponding to the 3rd quartile, $n$ = 2679), scaled and centered (referred to as "scaled cpm" values). Principal component analysis (PCA) was performed on scaled cpm values of the most variable genes expressed in both B cell and MEF reprogramming. The groups shown were obtained by $k$-means clustering. For heatmaps representing scaled PSI and scaled cpm values in B cell and MEF reprogramming, hierarchical clustering was performed on values of both datasets using Ward's method and Euclidean distance as the distance metric.

### Correlation of RNA-binding proteins expression to AS clusters

The lists of *Mus musculus* RNA-binding proteins (RBPs) and splicing-related proteins were downloaded from the *Uniprot* database. For RBPs, Uniprot keywords had to match "RNA-binding [KW-0694]," "mRNA splicing [KW-0508]," "mRNA processing [KW-0507]," or "Spliceosome [KW-0747]." Splicing-related RBPs were defined if keywords were matching either "mRNA splicing [KW-0508]" or "Spliceosome [KW- 0747]."

Filtered gene expression values were scaled as described above and a *membership* value was attributed to each RBP profiles for each AS cluster centroid and to its negative (RBPs with membership ≥ 0.3 are shown).

### Sequence feature analysis and RNA maps

Sequence features of exons and flanking introns were analyzed using *Matt* software v1.3.0 [39]. Features of TIA1-, CPSF3-, or UL1-dependent exons were compared with the corresponding independent exons and with a set of control cassette exons, representing alternative exons not changing during reprogramming. Specifically, the union of alternative non-changing exons (AS_NC sets, 10 < average PSI < 90 and ΔPSI ≤ 2) in all conditions of each dataset was generated, TIA1- or CPSF3/UL1-dependent events were excluded and a random set was selected, with a size similar to TIA1- or CPSF3-/UL1-dependent and TIA1- or CPSF3-/UL1-independent exon sets. Maximum entropy score is calculated as an approximation of splice site strength [73]. RNA maps for TIA1 M075 motif [74] were generated for the first and last 50 nt of exons and the first and last 150 nt of introns (sliding window = 31, $p$ value ≤ 0.05 with 1000 permutations).

### Statistical analyses and plots

Statistical tests were performed as indicated in figures legends with R (v3.6.1) or Graph-Pad Prism (version 8). Heatmaps were plotted with *ggplot2* package or *heatmap.2* function. Venn diagrams were generated with *VennDiagram* package [75].

### Gene Ontology enrichment analysis

Genes bearing the differentially spliced exons belonging to each set were analyzed for GO term enrichment in contrast to all genes containing any mapped AS event. Analysis was carried out with the "two unranked lists of genes" module of *GOrilla* (Gene Ontology enRIchment anaLysis and visuaLizAtion) tool [41] and summarized using *REViGO* [76]. Statistical significance was defined with $p$ value < 1e−03. Only enriched terms for GO process are shown.

### Alternative polyadenylation analysis

Transcript quantification was performed with *Salmon* v1.1.0 [77]. *QAPA* v1.3.0 was used to quantify poly-A site usage for each sample of the knockdown reprogramming dataset, based on the combined annotation of GENCODE and PolyAsite databases as described in [42]. A value of poly-A usage (PAU) was obtained for each site and filtering was applied as described in [42]: transcripts were selected for minimum expression (3 transcripts per million (tpm) in at least 10 out of 12 samples) and for number of poly-A sites (at least 2). Similarly to what was performed for AS, CPSF3- or UL1-dependent APA events were poly-A sites for which in both shRNAs |ΔΔPAU| ≥ 10, where ΔΔPAU = (ΔPAU shC/U#1/2_day12 − NI_day0) − (ΔPAU shSCR_day12 − NI_day0). Overlaps between the coordinates of AS events (largest junction) and APA events (last exon coordinates) were calculated with *Bedtools* v2.25.0 [78].

## Cell culture

Platinum E cells (PlatE), 293 T/17 cells, stromal S17 cells, and primary MEFs serving as feeder cells for B cell reprogramming were kindly provided by the group of T. Graf and cultured in Glutamax Dulbecco's modified Eagle's medium (DMEM, Life Technologies) supplemented with 10% fetal bovine serum (FBS) and penicillin/streptomycin antibiotics (PenStrep, 50 U/ml penicillin; 50 μg/ml streptomycin). All cells were maintained at 37 °C under 5% $CO_2$ and routinely tested for mycoplasma.

## Lentivirus production

293T/17 cells were seeded on gelatin-coated 10-cm plates at a density of $3 \times 10^6$ cells/plate and incubated overnight. The following day, medium was replaced approximately 2 h before transfection. For each 10-cm plate, 10 μg of the plasmid of interest, 2.5 μg of VSV-G, and 7.5 μg of pΔ8.9 were mixed with 61 μl of CaCl2 2.5 M (Sigma) and endotoxin-free water up to 500 μl. While bubbling, 500 μl of 2× HBS pH 7.2 (281 mM NaCl, 100 mM HEPES, 1.5 mM Na2HPO4) were added dropwise, and the mix was incubated for 10 min at room temperature (RT). The solution was added dropwise to the 293T/17 cells, followed by incubation for 24 h at 37 °C. The following day, medium was substituted with 6 ml of fresh medium (reprogramming medium for MEF reprogramming). Viral medium was collected 48 and 72 h after transfection, filtered (0.22 μm pore size), and supplemented as needed.

## Retrovirus production

$4.5 \times 10^6$ PlatE cells were seeded in gelatin-coated 10-cm plates (0.1% gelatin, Millipore) the day before transfection. To improve efficiency, chloroquine was added approximately 1 h before transfection to a final concentration of 30 μM. For each plate, 20 μg of plasmid was mixed with 60 μl of CaCl2 2.5 M (Sigma) and endotoxin-free water up to 500 μl. While bubbling, 500 μl of 2× HBS pH 7.2 (281 mM NaCl, 100 mM HEPES, 1.5 mM Na2HPO4) were added dropwise, and the mix was incubated for 10 min at room temperature (RT). The solution was added dropwise to the PlatE cells and transfected cells were incubated for 8–10 h. Medium was substituted with 6 ml of fresh medium (reprogramming medium for MEF reprogramming). Viral medium was collected the day after transfection and the following one, filtered (0.22 μm filter pore size), and supplemented as needed.

## C/EBPα-dependent B cell reprogramming

Primary B cell reprogramming was performed as described in [18–20, 79].

## MEF reprogramming

Primary MEFs (P0), obtained from male embryos of a *Collagen-OKSM, M2rtta⁺,* mouse line [30] were cultured on gelatin-coated plates in MEF medium (DMEM; high glucose + Glutamax, FBS 10%, PenStrep 1×, Sodium Pyruvate 1 mM, HEPES 30 mM, NEAA 1x, β-mercaptoethanol 0.1 mM). Primary MEFs were expanded for a maximum of 2 passages before inducing reprogramming. Early passage MEFs were plated in MEF medium on gelatin-coated 6-well plates at a density of 70,000 cells/well. The following day, infection was performed by substituting the medium with

the filtered retroviral or lentiviral supernatant (prepared in reprogramming medium), supplemented with LIF and doxycycline to induce reprogramming. Two subsequent infections were performed, 12 h apart, after which fresh reprogramming medium was added. The following day (day 2), cells were trypsinized and FACS-sorted for GFP expression by flow cytometry. Sorted MEFs were seeded (10,000 cells/well of 12-well plate) on irradiated feeders on gelatin-coated plates (plated the previous day 100,000 cells/well of 12-well plate). Medium was substituted every 2 days starting from day 4. Harvesting was performed every 2 days with trypsin 0.25% to ensure the complete dissociation of the feeder layer. Doxycycline was withdrawn from the culture at day 12 post-OSKM induction and AP staining was performed at day 14 with the Alkaline Phosphatase Kit II (StemGent), according to the manufacturer's instructions. The number of AP$^+$ colonies was quantified with ImageJ software (colony size between 20 and 2000 pixels).

Reprogramming medium is composed by DMEM, high glucose + Glutamax, ES-qualified FBS 15%, PenStrep 0.5×, Sodium Pyruvate 1 mM, HEPES 30 mM, NEAA 1×, and β-mercaptoethanol 0.1 mM. LIF (1000 U/ml) and doxycycline (1 μg/ml) were added freshly.

### Flow cytometry analysis

Cells were trypsinized and stained with a mix of antibodies SSEA1-eFluor 660 (MC-480, eBioscience) and EpCAM-PE (G8.8, eBioscience), both at a 1:50 dilution in 100 μl per sample. Staining was carried out by incubating on ice for 20 min, followed by washing and staining with DAPI (Sigma). Flow cytometry analysis was performed with a BD LSR II analyzer. Gates and compensation between FITC and PE were adjusted on non-stained controls. In total, 10,000 events (alive cells) per sample were acquired.

### Cloning of LEF1 isoforms and TIA1 in retroviral vectors

*Lef1* isoforms were amplified from bulk cDNAs of B cells and iPS (RNA-seq samples) and cloned using Gibson technology [80] into a MIG-pMSCV retroviral vector including an IRES-GFP element. Kozak consensus sequences and an N-terminal T7 tag were inserted before the LEF1 ORF to ensure efficient translation and discrimination from endogenous *Lef1*.

cDNA of mouse *Tia1* was purchased from GenScript in pcDNA3.1 vectors (clone ID: OMu08423D) and cloned using a similar Gibson strategy into the MIG-pMSCV retroviral vector bearing the N-terminal T7 and IRES-GFP elements. Gibson reaction master mixes were provided by the CRG Protein Technologies Core Facility.

### Short hairpin RNAs

Five MISSION shRNAs specific for each splicing regulator were purchased from Sigma in pLKO lentiviral vectors, and their effects were compared with those of SHC002 mammalian non-targeting MISSION shRNA (Sigma). The effect of each shRNA was tested on MEFs and E14 embryonic stem cells, and knockdown efficiency was assessed by RT-qPCR at 72/96 h post-infection using primers specific for each target and normalized by the expression of two housekeeping genes

(*Gapdh* and *Rplp0*). The two shRNAs displaying the largest knockdown effects in both cell lines were selected (sequences shown in Additional file 7: Table S6).

### RNA extraction and retrotranscription, semi-quantitative RT-PCR, and real-time qPCR (RT-qPCR)

RNA extraction and DNAse treatment was performed using Maxwell simplyRNA kit (Promega) or RNeasy Mini kit (Qiagen), following the manufacturers' instructions. In total, 200 ng of total RNA were reverse-transcribed with Superscript III (Invitrogen) following the manufacturer's recommendations.

PCR reactions were carried out using GoTaq enzyme (Promega) with 1 µl of cDNA diluted 1:5. To quantify inclusion of alternatively spliced exons, capillary electrophoresis was performed using a Labchip GX Caliper workstation (Caliper, Perkin Elmer) at the CRG Protein Technologies Core Facility. The nanomolar content of each band was extracted with LabChip GX software and PSI values were calculated as the ratio between the inclusion amplicon and the sum of inclusion and skipping amplicons.

Real-time quantitative PCR (RT-qPCR) was performed on a ViiA7 Real Time PCR System (Applied Biosystems). Reactions in a total volume of 10 µl contained 2× SYBR Green Master Mix (Applied Biosystems), primers 400 nM and 1 µl of previously synthetized cDNA, diluted 1:5–20. The output Ct values were normalized by the expression of the housekeeping gene *Gapdh* (unless differently stated) and analyzed with ΔCt/ΔΔCt method. All primers sequences are listed in Additional file 7: Table S6.

### Supplementary Information

---

**Additional file 1.** Supplementary figures and corresponding legends.

**Additional file 2: Table S1.** Cassette exon events differentially spliced in B cell reprogramming. This table includes the information about the AS cluster each exon belongs to and the predicted effect on its transcript. Data for other types of events is available upon request.

**Additional file 3: Table S2.** Cassette exon events differentially spliced in MEF reprogramming. This table includes the information about the overlap with B cell reprogramming. Data for other types of events is available upon request.

**Additional file 4: Table S3.** Gene expression levels of RBPs correlating with AS clusters (positive and negative predicted regulators).

**Additional file 5: Table S4.** CPSF3-, UL1- and TIA1-dependent and -independent events. This table includes the sets of control cassette exons (CEx) used for KD (CPSF3- and UL1-dependent CEx) or OE (TIA1-dependent CEx).

**Additional file 6: Table S5.** Gene Ontology terms enriched in CPSF3-, UL1- and TIA1-dependent events. This table includes outputs from GOrilla and REViGO analyses.

**Additional file 7: Table S6.** Sequences of primers and shRNAs used in this study.

**Additional file 8.** Review history.

---

#### Acknowledgements
We thank the CRG Genomics Unit for sequencing; the CRG Tissue Engineering Unit for the inactivated MEF feeders; the CRG/UPF FACS Unit for FACS sorting; Daniel Soronellas for help with the *Mfuzz* package and Konrad Hochedlinger for the *Col-OKSM* reprogrammable mouse strain. We thank members of Manuel Irimia's, T.G.'s and J.V.'s groups for discussions; Manuel Irimia and members of J.V.'s group for critical reading of the manuscript.

#### Review history
The review history is available as Additional file 8.

#### Peer review information

## Authors' contributions
C.V., T.G., and J.V. conceived the study. C.V., P.P., J.L.S., and J.V. designed experiments and data analyses. C.V. and J.V. wrote the manuscript with input from all authors. C.V. performed the majority of the experimental work and data analyses (RNA-seq analyses, MEF reprogramming, and molecular biology experiments). R.S., B.D.S., and C.B. performed the B cell reprogramming experiments for RNA-seq. A.R.R. performed some RT-PCR validations and provided technical support. S.G, A.M. and B.P. provided reprogrammable MEFs. J.V. and T.G. acquired funding. The authors read and approved the final manuscript.

## Authors' information
Twitter handles: @clau_vivori (Claudia Vivori); @R_Stadhouders (Ralph Stadhouders); @AnnaRR82 (Anna Ribó Rubio); @AnMallol (Anna Mallol); @Payerlab (Bernhard Payer); @JuanValcarcel5 (Juan Valcárcel).

## Funding
C.V. was recipient of an FPI-Severo Ochoa Fellowship from the Spanish Ministry of Economy and Competitiveness. Work in J.V. laboratory is supported by the European Research Council (ERC AdvG 670146), AGAUR, Spanish Ministry of Economy and Competitiveness (BFU 2017 89308-P) and the Centre of Excellence Severo Ochoa. Work in T.G.'s laboratory was supported by the European Research Council FP7/2007-2013 (ERC Synergy Grant 4D-Genome) the Ministerio de Educación y Ciencia (SAF.2012-37167) and AGAUR. We acknowledge support of the Spanish Ministry of Science and Innovation to the EMBL partnership and the CERCA Programme / Generalitat de Catalunya.

## Availability of data and materials
RNA-seq data of MEF reprogramming upon TIA1 overexpression and knockdown of CPSF3 and hnRNP UL1, generated during the current study, have been deposited in NCBI's Gene Expression Omnibus [68] and are accessible through GEO Series accession number GSE158633 [81]. The code used to process and analyze these data was released under the GNU GPL-3 license (https://doi.org/10.5281/zenodo.4724548) and is available at https://github.com/cvivori/AS-reprogramming-KD-OE [82]. The RNA-seq data of B cell [20] and MEF reprogramming [17] analyzed in the current study are available under GEO Series accession numbers GSE96611 and GSE70022, respectively. The code used to process and analyze these data was released under the GNU GPL-3 license (https://doi.org/10.5281/zenodo.4723460) and is available at https://github.com/cvivori/AS-BandMEF-reprogramming [83].

# Declarations

## Ethics approval and consent to participate
Not applicable.

## Competing interests
The authors declare no competing interests.

## Author details
[1]Centre for Genomic Regulation (CRG), The Barcelona Institute of Science and Technology, Carrer del Dr. Aiguader 88, 08003 Barcelona, Spain. [2]Universitat Pompeu Fabra (UPF), Carrer del Dr. Aiguader 88, 08003 Barcelona, Spain. [3]Present address: The Francis Crick Institute, 1 Midland Road, London NW1 1AT, UK. [4]Present address: Friedrich Miescher Institute for Biomedical Research, Maulbeerstrasse 66/Swiss Institute of Bioinformatics, 4058 Basel, Switzerland. [5]Present address: Departments of Pulmonary Medicine and Cell Biology, Erasmus MC, Rotterdam, The Netherlands. [6]Present address: Department of Molecular and Cellular Biology, Baylor College of Medicine, One Baylor Plaza, Alkek Bldg Room N1020, Houston, TX 77030, USA. [7]Present address: Josep Carreras Leukaemia Research Institute, Carretera de Can Ruti, Camí de les Escoles, s/n, 08916 Badalona, Spain. [8]Institució Catalana de Recerca i Estudis Avançats (ICREA), Passeig Lluís Companys 23, 08010 Barcelona, Spain.

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

## 
