## [**Additional file 8.** Review history. · Genome Biology]

Review History

First round of review

Reviewer 1

Are you able to assess all statistics in the manuscript, including the appropriateness of statistical tests used? Yes, and I have assessed the statistics in my report.

Comments to author:

Using somatic cells to generate iPS hold a great promise in personal medicine. This is the first manuscript that studies changes in the splicing program during such a reprogramming process. In this manuscript Vivori et al., used a very elegant assay to generate pluripotent cells from mouse B cell and MEF cells and then analyzed splicing transcriptome-wide. The results provide deep understanding on these reprogramming systems during the transitions to pluripotency, identifying thousands of exons and introns that are involved in this process. Importantly, they demonstrated that during early stages in the reprogramming process splicing events disrupt open reading frames, whereas in late stages of the reprogramming process the splicing events maintain the open reading frames. Furthermore, they identified many splicing factors and RNA binding proteins of previously unknown functions that are involved in the transitions to pluripotency. They selected three of these splicing factors (CPSF3, hnRNP UL1, and TIA1) for further analyses, and demonstrated their involvement in the transitions to pluripotency in MEF cells.

The technical difficulties involved in manipulation of the B cells (page 10 onward) and the need to do it on the MEF cells hampered the flow of the manuscript. Nevertheless, the similar trends in both B cell and MEF cells in the reprogramming process indicate that the results are general.

Overall, the manuscript is well written and the results are novel and important to a wide range of disciplines. I therefore support publishing this manuscript in its current form with only minor changes.

Minor points

1. Although it is common knowledge, I recommend that a paragraph be added to the Discussion to explain the technical difficulties involved in manipulation of B cells.
2. The first time that the activation of MEFs appears in the text is at the end of the introduction and not at the beginning of the Results section - please correct this.

Reviewer 2

Are you able to assess all statistics in the manuscript, including the appropriateness of statistical tests used? Yes, and I have assessed the statistics in my report.

Comments to author:

This manuscript by Valcarcel et al. analyzed AS events in reprogramming of B cell and MEF to iPS cells. They specifically examined CPSF3, hnRNP UL1 and TIA1 for their roles in B cell to iPS cell reprogramming. Overall the work was well carried out. There are however

several major issues the authors need to address before the work can be accepted for publication.

Major:

1. CPSF3' role in AS regulation is novel, at least in mammalian cells. It is unclear if its role in cell reprogramming is direct or indirect. It is possible that some of the AS events may well be caused by inhibition of reprogramming and have nothing to do with CPSF3. In other words, any inhibitor of reprogramming, or perhaps even just cell proliferation, may have similar effects on these AS events. This is a key issue for this paper which must be addressed. One possible experiment is to OE CPSF3 and examine its effect on AS and reprogramming, like what they did with TIA1. ChIP-seq analysis of CPSF3 binding near splice sites would also be informative.

Minor:

1. Figure 1H. The numbers in each bar are too small to be legible. Maybe a table shown next to the plot could help.
2. The authors should check other 3' end processing factors. The ones shown in Fig. 2E are complete.
3. It is not clear how significant is the correlation between CPSF3, hnRNP UL1 or TIA1 and AS clusters (Fig. 3). The authors should calculate correlations for all RBPs and then indicate where these three factors are positioned. This should give readers a perspective as to why these factors were isolated for further investigation.
4. Because CPSF3 is a 3' end processing factor, the authors may want to check alternative polyadenylation as well, using the same RNA-seq data. There are several programs that do this type of analysis.

Vivori et al (GBIO-D-20-01977) resubmission

Response to reviewers

Reviewer #1: ==Using somatic cells to generate iPS hold a great promise in personal medicine. This is the first manuscript that studies changes in the splicing program during such a reprogramming process. In this manuscript Vivori et al., used a very elegant assay to generate pluripotent cells from mouse B cell and MEF cells and then analyzed splicing transcriptome-wide. The results provide deep understanding on these reprogramming systems during the transitions to pluripotency, identifying thousands of exons and introns that are involved in this process. Importantly, they demonstrated that during early stages in the reprogramming process splicing events disrupt open reading frames, whereas in late stages of the reprogramming process the splicing events maintain the open reading frames. Furthermore, they identified many splicing factors and RNA binding proteins of previously unknown functions that are involved in the transitions to pluripotency. They selected three of these splicing factors (CPSF3, hnRNP UL1, and TIA1) for further analyses, and demonstrated their involvement in the transitions to pluripotency in MEF cells.

The technical difficulties involved in manipulation of the B cells (page 10 onward) and the need to do it on the MEF cells hampered the flow of the manuscript. Nevertheless, the similar trends in both B cell and MEF cells in the reprogramming process indicate that the results are general.

Overall, the manuscript is well written and the results are novel and important to a wide range of disciplines. I therefore support publishing this manuscript in its current form with only minor changes.

We very much appreciate the reviewer's positive comments and support.

Minor points

1. Although it is common knowledge, I recommend that a paragraph be added to the Discussion to explain the technical difficulties involved in manipulation of B cells.

A paragraph has been added to the Discussion along the lines requested by the reviewer (p. 18).

2. The first time that the activation of MEFs appears in the text is at the end of the introduction and not at the beginning of the Results section - please correct this.

The definition of the MEFs acronym is now made the first time that the term appears in the text (line 28, p4) instead of at the beginning of the Results section. Thanks a lot!

Reviewer #2: This manuscript by Valcarcel et al. analyzed AS events in reprogramming of B cell and MEF to iPS cells. They specifically examined CPSF3, hnRNP UL1 and TIA1 for their roles in B cell to iPS cell reprogramming. Overall the work was well carried out. There are however several major issues the authors need to address before the work can be accepted for publication.

We appreciate the reviewer's positive overall evaluation of our work and thank him/her for the constructive comments.

Major:

1. CPSF3' role in AS regulation is novel, at least in mammalian cells. It is unclear if its role in cell reprogramming is direct or indirect. It is possible that some of the AS events may well be caused by inhibition of reprogramming and have nothing to do with CPSF3. In other words, any inhibitor of reprogramming, or perhaps even just cell proliferation, may have similar effects on these AS events. This is a key issue for this paper which must be addressed. One possible experiment is to OE CPSF3 and examine its effect on AS and reprogramming, like what they did with TIA1. ChIP-seq analysis of CPSF3 binding near splice sites would also be informative.

Thank you for bringing this up. One first point to comment on is that work in Michael Green's lab previously reported global promotion of alternative internal exon usage by mRNA 3' end formation factors CPSF/SYMPK (Misra et al, Mol Cell 2015, Genomics Data 2015; RNA Biol 2016), which we now discuss to larger extent in the text (pp. 10 and 18-19). Nevertheless, the reviewer makes a valid point that the effects of CPSF3 on reprogramming cannot rigorously be directly associated with the observed changes in AS. We would like to point out, however, that 1) as stated on p.12, survival of cells during reprogramming did not seem to be affected in the knockdowns because no significant difference in the proportion of DAPI-stained cells was observed throughout reprogramming in any condition (Figure S4D), and 2) not all genetic perturbations leading to inhibition of reprogramming are associated with the same program of AS changes. For example, our own data indicate that while knockdown of either CPSF3 or hnRNP UL1 lead to a highly overlapping set of AS changes, overexpression of TIA1, also leading to inhibition of reprogramming, is associated with a largely non-overlapping set of AS changes. We discuss this further in the text (p 19).

We have followed the reviewer's suggestion to overexpress CPSF3 and examined the effects on the expression of pluripotency markers, reprogramming efficiency and key AS events. The following Figures summarize the results:

Figure 1. Effects of CPSF3 overexpression on the expression of pluripotency markers and MEFs reprogramming. (A) Expression levels of *Cpsf3* mRNA relative to *Gapdh*, quantified by RT-qPCR in non-infected cells (NI), cells transduced with an empty vector (Empty) or with T7-*Cpsf3* cDNA. (B) Expression levels of *Pou5f1* and *Nanog* relative to *Gapdh*, quantified by RT-qPCR as in panel A. (C) Percentage of SSEA1⁺EPCAM1⁻ early reprogramming intermediates (day 6 post-OSKM induction) upon T7-*Cpsf3* overexpression determined by flow cytometry. (D) Percentage of SSEA1⁺EPCAM1⁺ late reprogramming intermediates (days 10 and 12 post-OSKM induction) upon T7-*Cpsf3* overexpression. (E) Number of alkaline phosphatase (AP) positive colonies at day 14 post-OSKM induction upon T7-*Cpsf3* overexpression. Images of representative wells are shown below.

Figure 1 (now also part of Figure S4) shows that, contrary to the reduction of OCT4 (*Pou5f1*) and NANOG expression observed upon CPSF3 knockdown (Figure 4E of the manuscript), CPSF3 overexpression resulted in a trend towards increased expression of these factors at late times of reprogramming (panel A). However this seems to be insufficient for enhanced reprogramming efficiency or redistribution of reprogramming intermediates (panels B-D). This is likely due to additional rate-limiting steps required to increase the efficiency of this complex process, as well as to experimental limitations related to the timing and levels of overexpression achieved in the experiment, including for example stoichiometric unbalances with co-factors (e.g. Misra et al, 2015 observed that CPSF/SYMPK

acted in concert with RNA binding proteins, see also below). As pointed out by others (Liu et al Nat Cell Biol 15:829 (2013) and Zavolan & Kanitz Curr Op Cell Biol 52:8 (2018)), optimal reprogramming is highly influenced by time-sensitive requirements for individual factors.

We selected a number of alternative splicing events affected by CPSF3 knockdown (which, given the strong overlap between their splicing programs, were also affected by hnRNP UL1 knockdown) and tested the effects of CPSF3 overexpression on these events. Figure 2 (now also Figure S6F) illustrates the more limited changes in alternative splicing during reprogramming observed upon knockdown of CPSF3 or hnRNP UL1 (measured by RT-PCR assays that confirm the changes predicted by RNA-seq analyses).

Figure 2. Impact of the knockdown of CPSF3 or hnRNP UL1 on changes in alternative splicing observed upon MEFs reprogramming. (F) Inclusion levels of selected CPSF3- and UL1-dependent AS events, quantified by semi-quantitative RT-PCR and capillary electrophoresis. Values represent average and SD.

Figure 3 shows that overexpression of CPSF3 had no clear effect on these alternative splicing changes. As mentioned above, the final model proposed by Misra et al (2015) for the function of CPSF/SYMPK in alternative splicing was that these factors are recruited to their targets RNAs via other RNA Binding Proteins such as RBFOX2, NOVA2 or hnRNP A1. Similarly, overexpression of CPSF3 may not lead to general changes in alternative splicing in the absence of elevated levels of its cofactors, which may be at the basis of the limited effects of CPSF3 overexpression on cell reprogramming, affecting the expression of targets such as OCT4 and NANOG, but not of other genes / pathways required for reprogramming.

Figure 3. Impact of CPSF3 overexpression on changes in alternative splicing observed upon MEFs reprogramming. Values represent average and SD.

MEFs reprogramming. Inclusion levels of selected CPSF3- and UL1-dependent AS events, quantified by semi-quantitative RT-PCR and capillary electrophoresis. Individual values and their averages are represented.

Given the limitations of these results, we have included in the manuscript (Supplementary Figure S4) the results of Figure 1 above illustrating the reverse effects of CPSF3 knockdown and overexpression on OCT4 and NANOG expression and the absence of effects on reprogramming efficiency, referring in the text to the absence of effects of overexpression on alternative splicing, as well as incomplete impact on the complex process of reprogramming, as likely causes for the absence of improved reprogramming.

Regarding the final suggestion of the reviewer: given the lack of published mouse CPSF3 CLIP data or of a described motif, we could not at this point infer binding of CPSF3 around its target exons. De novo motif enrichment analyses did not reveal either any particular sequence enriched around CPSF3-dependent exons compared to the control set of alternative exons. As discussed above, if the model proposed by Misra et al (2015) for the function of CPSF/SYMPK in alternative splicing applies to CPSF3, recruitment of CPSF3 to their targets via other RNA Binding Proteins could explain our failure to detect a well-defined unique motif.

Minor:

1. Figure 1H. The numbers in each bar are too small to be legible. Maybe a table shown next to the plot could help.

We have increased the numbers sizes and indicated in the figure legend that percentages lower than 5% are not shown, as these represent only minor categories.

2. The authors should check other 3' end processing factors. The ones shown in Fig. 2E are complete.

We have now further clarified in the text (pp 8-9) that we initially examined an unbiased collection of 214 RBPs, of which those represented in Figure 2E were those displaying changes in expression among embryonic stem cells and differentiated tissues according to Han et al., 2013. These included some components of the 3' end processing machinery, but not all. Similarly, only a fraction of splicing factors are represented in Figure 2E: only those that similarly showed changes in expression between the pluripotent and differentiated compartments in Han et al., 2013. In other words, the heatmap illustrates the similarities and differences between expression patterns of these pluripotency-associated RBPs in the B cell and MEF reprogramming datasets.

A larger set of 507 RBPs (annotated as RNA-binding proteins by the Uniprot

database) was instead considered to infer potential regulators of our clusters of alternatively spliced exons (Figure 3).

3. It is not clear how significant is the correlation between CPSF3, hnRNP UL1 or TIA1 and AS clusters (Fig. 3). The authors should calculate correlations for all RBPs and then indicate where these three factors are positioned. This should give readers a perspective as to why these factors were isolated for further investigation.

Measurement of the correlations between clusters of alternative splicing changes and expression profiles of RBPs (membership value attributed by the *Mfuzz* package) is provided in Supplementary Table S3. The choice of splicing factors selected for further investigation was partially guided by these correlations and their pattern of expression in both B cell and MEFs reprogramming, and partially by other considerations, including evidence (or lack of) in the literature regarding their role in development/pluripotency or alternative splicing regulation. This is now better explained in the text (p. 10).

4. Because CPSF3 is a 3' end processing factor, the authors may want to check alternative polyadenylation as well, using the same RNA-seq data. There are several programs that do this type of analysis.

Thanks a lot for this suggestion. Using *Matt*, we had analyzed the relative position of the CPSF3-dependent AS exons in their transcripts compared to CPSF3-independent and control AS exons, and we had found no enrichment in last exons in these sets compared to controls. We have now run the 'Quantification of APA' (QAPA) pipeline (Ha, Blencowe and Morris, 2018; Ha *et al.*, 2021) on our RNA-seq datasets to quantify alternative poly-A usage. Not unexpectedly, we do observe changes in 3' processing sites by knockdown of CPSF3 (and interestingly also by knockdown of hnRNP UL1), although the overlap between CPSF3- and UL1-dependent AS events and their corresponding APA regions is minimal (Figure S6G). Additionally, the set of genes regulated by CPSF3 and UL1 at the level of alternative splicing and alternative poly-A selection are essentially distinct (Figure S6H). This suggests – along the lines of our response to major point 1 above – that CPSF3 and other 3' end formation factors do have relevant functions in splicing regulation independent of their essential functions in 3' end cleavage and polyadenylation. These results have been included as part of Supplementary Figure S6 and discussed in the text (p. 15), including an explicit reference to the possibility that the contributions of these factors to cell reprogramming is based upon their activities as splicing regulators, polyadenylation regulators, or both.

Once again, we thank this reviewer for the constructive suggestions, which have clearly helped to improve the manuscript.

Second round of review

Reviewer 2

The authors have addressed all my concerns.